# Does Object Binding Naturally Emerge in Large Pretrained Vision Transformers?

Yihao Li[1]    Saeed Salehi[2]    Lyle Ungar[1]    Konrad P. Kording[1]

[1]University of Pennsylvania    [2]Machine Learning Group, Technical University of Berlin

liyihao@seas.upenn.edu, ai.neuro.io@gmail.com
ungar@cis.upenn.edu, koerding@gmail.com

## Abstract

Object binding, the brain's ability to bind the many features that collectively represent an object into a coherent whole, is central to human cognition. It groups low-level perceptual features into high-level object representations, stores those objects efficiently and compositionally in memory, and supports human reasoning about individual object instances. While prior work often imposes object-centric attention (e.g., Slot Attention) explicitly to probe these benefits, it remains unclear whether this ability naturally emerges in pre-trained Vision Transformers (ViTs). Intuitively, they could: recognizing which patches belong to the same object should be useful for downstream prediction and thus guide attention. Motivated by the quadratic nature of self-attention, we hypothesize that ViTs represent whether two patches belong to the same object, a property we term *IsSameObject*. We decode *IsSameObject* from patch embeddings across ViT layers using a quadratic similarity probe, which reaches over 90% accuracy. Crucially, this object-binding capability emerges reliably in DINO, CLIP, and ImageNet-supervised ViTs, but is markedly weaker in MAE, suggesting that binding is not a trivial architectural artifact, but an ability acquired through specific pretraining objectives. We further discover that *IsSameObject* is encoded in a low-dimensional subspace on top of object features, and that this signal actively guides attention. Ablating *IsSameObject* from model activations degrades downstream performance and works against the learning objective, implying that emergent object binding naturally serves the pretraining objective. Our findings challenge the view that ViTs lack object binding and highlight how symbolic knowledge of "which parts belong together" emerges naturally in a connectionist system. [1]

## 1  Introduction

Humans naturally parse scenes into coherent objects [1] (e.g., grouping features such as rounded shape, smooth surface, and muted color into *the mug*) and further ground their identities in context (e.g., recognizing *my coffee mug on the desk* rather than just *a mug*). This is assumed to be made possible by what cognitive scientists call *object binding* [2], the brain's ability to group an object's low-level features (color, shape, motion, etc.) into a unified representation. This in turn enables objects to be stored efficiently and compositionally in memory and used as high-level symbols for reasoning. The binding problem is a genuine computational challenge, as evidenced by humans' limited competence in conjunction-search tasks [3] and clinical dissociations such as Balint's syndrome, where feature perception remains intact but binding breaks down [4]. If AI systems could replicate the human ability for object binding, that may help them ground symbols for

---

[1]Code available at: https://github.com/liyihao0302/vit-object-binding.

39th Conference on Neural Information Processing Systems (NeurIPS 2025).

perception and exploiting compositionality [5]. The key question is: do current AI systems solve the binding problem?

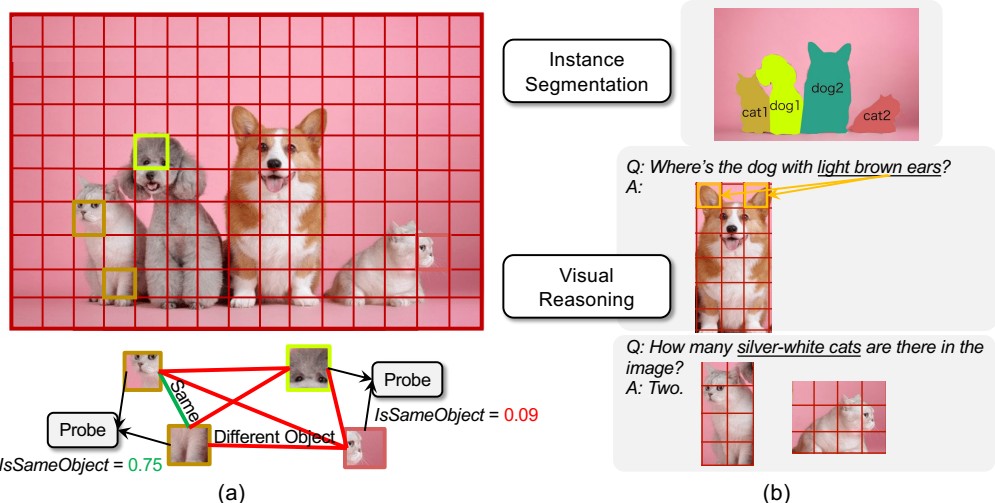

Figure 1: **Assessing object binding in ViTs with *IsSameObject*.** (a) We use a probe to decode *IsSameObject*, with scores near 1 for same-object pairs and near 0 for different-object pairs. (b) Downstream tasks that benefit from strong object binding include instance segmentation and visual reasoning (e.g., locating and counting objects with specific features), where patches triggered by certain features are bound to the rest of their object to allow extraction of the entire object.

Object binding has received little attention in mainstream AI research. Cognition-inspired models [6, 7] build in human-like object-based attention. By contrast, mainstream vision models are assumed to implicitly learn to handle multiple objects from training data, yet empirical studies show they often "attend" only to the most salient regions and overlook the rest [8]. While ViT attention scores can capture global image structure and salient regions (often corresponding to target objects) [9], empirical evidence shows that self-attention tends to group patches by low-level feature similarity rather than reliably producing object-level binding [10]. Object-centric methods like Slot Attention [11] fix this by allocating a small set of learnable slots that compete for token features, enforcing binding by design. However, whether AI vision models, especially leading ViTs, can achieve robust object binding without *explicit* mechanisms remains an open question.

Cognitive scientists have questioned whether ViTs can bind objects at all: arguing that they lack mechanisms for dynamically and flexibly grouping features [5]; they lack recurrence necessary for iterative refinement of object representations [12, 7]; and as purely connectionist models, they appear incapable of true symbolic processing [8]. However, these architectural limitations do not preclude binding from emerging through learning. If a model encodes whether two patches belong to the same object (*IsSameObject*), this signal can guide attention and improve prediction [9, 13]. Human-labeled data also reflects object-level structure, so ViTs can acquire binding by imitation. This suggests that ViTs may learn to bind objects directly from large-scale training data, without requiring explicit architectural inductive biases.

Here, we ask whether object binding naturally emerges in large, pretrained Vision Transformers, which is a question that matters for both cognitive science and AI. We propose *IsSameObject* (whether two patches belong to the same object) and show that it is reliably decodable (with 90.20% accuracy) using a quadratic similarity probe starting from mid-layers of the transformer layers. This effect is robust across DINO, CLIP and ImageNet-supervised ViTs, but largely absent in MAE, suggesting that binding is an acquired ability rather than a trivial architectural artifact. Across the ViT's layer hierarchy, it progressively encodes *IsSameObject* in a low-dimensional projection-space on top of the features of the object, and it guides self-attention. Ablating *IsSameObject* from model activations hurts downstream performance and works against the pretraining objective.

Our main contributions are as follows: (i) We demonstrate that object binding naturally emerges in large, pretrained Vision Transformers, challenging the cognitive-science assumption that such binding isn't possible given their architecture. (ii) We show that ViTs encode a low-dimensional signature of *IsSameObject* (whether two patches belong to the same object) on top of their feature representations. (iii) We suggest that learning-objective–based inductive biases can enable object binding, pointing future work toward implicitly learned object-based representations.

## 2 Related Work

**Object Binding in Cognitive Science and Neuroscience.** The object binding problem asks how the brain integrates features that are processed across many distinct cortical areas into coherent object representations [14]. The concept of binding[2] rests on three key hypotheses: First, visual processing is widely understood to be hierarchical, parallel, and distributed across the cortex [16–20]. Second, we perceive the world primarily in terms of objects, rather than as a collection of scattered features [12, 1]. This abstraction is fundamental to both perception and interaction with the world, allowing us to recognize, reason about, and manipulate our environment effectively [21–23]. Third, feature binding requires a mechanism that correctly assigns features, represented in spatially distinct cortical areas, to their corresponding object [15, 24, 2]. This third hypothesis is where the core of the binding problem lies, and it has been a longstanding point of debate among neuroscientists and cognitive scientists [25–27].

Despite their substantial difference, vision transformers (ViTs) share several key computational parallels with the mammalian visual system: they both rely on parallel, distributed and hierarchical processes. More importantly, ViTs do have two of the three architectural and computational elements hypothesized to enable binding in the brain. The explicit position embeddings in ViTs resemble spatial tagging and the spatiotopic organization observed in the ventral stream [28, 26]; and the self-attention mechanism is akin to dynamic tuning and attentional modulation, which are thought to be primary mechanisms for object binding [26, 29, 30] (although attention is believed to be of recurrent nature in the brain [31, 32]). These parallels position ViTs as potential computational models for exploring object binding in both artificial and biological systems.

**Object-Centric Learning.** Motivated by how humans naturally reason about individual objects, Object-Centric Learning (OCL [11]) aims to represent a scene as a composition of disentangled object representations. While segmentation only partitions an image into object masks, OCL goes further by encoding each object into its own representation [33]. Unsupervised approaches such as MONet [34], IODINE [33], and especially Slot Attention [11] encode scenes into a small, permutation-invariant set of "slots" that are iteratively refined, producing robust object representations on both synthetic [11, 35] and real-world data [36, 37] and enabling compositional generation and manipulation [38–40]. However, since Slot Attention is added as an external module rather than integrated into the transformer architecture, it introduces additional challenges for scaling and training [41]. Other explicit object-centric approaches include Tensor Product Representations [42] and Capsule Networks [43].

Instead of object-centric approaches that explicitly enforce object-level attention, we propose an alternative view that ViTs may already encode implicit object-level structure. Prior work has assumed this and attempted to group patches into objects directly from activations or attention maps ViTs, using methods like clustering [44] or GraphCut [45]. [46] conduct a behavioral experiment where participants judge whether two dots belong to the same object at varying distances, and show that patch-level feature similarity in self-supervised ViTs supports object-based grouping. Building on this line of work, we show that ViT patch embeddings intrinsically encode whether any two patches belong to the same object, and analyze how this information is structured through probing.

**Binding in Transformers.** Binding has received growing recognition in transformer-based machine learning research and binding failures are seen as examples of performance breakdowns in modern applications [47–50]. Diffusion models rely on binding attributes to entities, and failures cause attribute leakage (e.g., both a dog and a cat end up wearing sunglasses and a sun-hat) [48, 47]. Vision-language models face similar binding challenges, struggling with differentiating multiple objects with feature conjunctions [50]. Despite these binding failures, transformers still demonstrate some binding

---

[2]The term binding was introduced to neuroscience by Christoph von der Malsburg in 1981, inspired by the notion of variable binding in computer science [15].

capability, yet the underlying mechanism is not well understood. Feng and Steinhardt [51], Dai et al. [52] study binding in language models, showing that attributes (e.g., "lives in Shanghai") are linked to their subjects (e.g., "Alice") via a low-dimensional *binding-ID* code that is added to the activation and can be edited to swap or redirect relations. Binding mechanisms in vision transformers remain unexplored, and our study aims to fill this gap.

## 3 Assessing Object Binding in ViTs through *IsSameObject*

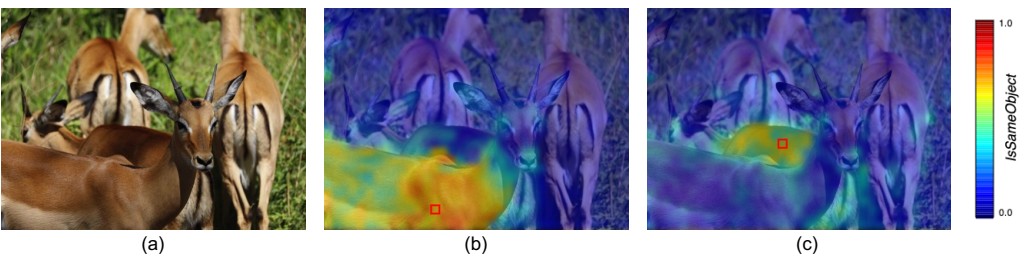

Figure 2: ***IsSameObject* predictions distinguish objects in highly complex scenarios [53].** Quadratic probe results for DINOv2-Large at layer 18 show that overlapping deer can be distinguished and that disconnected regions of the same deer are correctly retrieved in (c).

### 3.1 Probing *IsSameObject* representations

Vision Transformers (ViTs) tokenize images by dividing them into a grid of fixed-size patches [54]. Because the token is the minimal representational unit, any grouping of features into objects must arise through relations between tokens, not within them. The only mechanism ViTs have for such cross-token interaction is scaled dot-product attention, where attention scores can be viewed as dynamic edge weights in a graph that route information between tokens [5]. Therefore, if ViTs perform any form of object binding, we expect to observe a pairwise token-level representation that indicates whether two patches belong to the same object, which we term *IsSameObject*.

Since object binding is the ability to group an object's features together, decoding *IsSameObject* reliably from ViT patch embeddings would provide direct evidence of object binding (and its representation) in the model. We adopt probing, which takes measurements of ViT activations with lightweight classifiers [55], to determine whether *IsSameObject* is encoded or unrecoverable by simple operations.

Formally, we define the *IsSameObject* predicate on a pair of token embeddings $(x_i^{(\ell)}, x_j^{(\ell)})$ at layer $\ell$ by

$$\textbf{\textit{IsSameObject}}\big(x_i^{(\ell)}, x_j^{(\ell)}\big) = \phi\big(x_i^{(\ell)}, x_j^{(\ell)}\big), \quad \phi : \mathbb{R}^d \times \mathbb{R}^d \to [0, 1],$$

where $\phi$ scores the probability that tokens $i$ and $j$ belong to the same object.

Here, we ask whether models reliably encode *IsSameObject* and, if so, what mechanisms they use to do so. We consider the following hypotheses about how *IsSameObject* may be encoded in the model's activations:

- **It may be *linear*** (recoverable by a weighted sum of features) **or fundamentally *quadratic*** (recoverable only through pairwise feature interactions).
- **It is a *pairwise* relationship versus a *pointwise* mapping** (i.e. the model first maps each patch to a discrete object identity or class, then compares).
- **The model tells objects apart using only broad *class* labels or object *identities*–**i.e., it may rely on class-level recognition ("dog vs. chair") instead of explicitly binding pixels to objects, as class labels already encode a coarse notion of object identity.
- **The signal is stored in a few *specialized dimensions* versus *distributed* across many dimensions.** In the former case, binding information would be isolated to a small subset of

channels, while in the latter it would be encoded diffusely (e.g., as rotated combinations of features) such that no single dimension carries the signal on its own.

To test these hypotheses, we decode *IsSameObject* using several probe architectures, each parameterized by a learnable matrix $W$ and a scalar bias $b$. All probes are constructed to be *symmetric* in their inputs, reflecting the constraint *IsSameObject*$(x, y) =$ *IsSameObject*$(y, x)$. Throughout, $\sigma(\cdot)$ denotes the sigmoid function.

**1. Linear probe.**

$$\textit{IsSameObject}_{\text{lin}}(x, y) \; = \; \sigma(Wx + Wy + b), \qquad W \in \mathbb{R}^{1 \times d}, \; b \in \mathbb{R}.$$

**2. Diagonal quadratic probe (specialized dimensions).**

$$\textit{IsSameObject}_{\text{diag}}(x, y) \; = \; \sigma\big(x^\top W y + b\big),$$

where $W \in \mathbb{R}^{d \times d}$ is constrained to be diagonal, so the probe uses only $d$ parameters, each corresponding to a single feature dimension.

**3. Quadratic probe (distributed).**

$$\textit{IsSameObject}_{\text{quad}}(x, y) \; = \; \sigma\big(x^\top W_1^\top W_2 \, y + b\big),$$

where $W_1, W_2 \in \mathbb{R}^{k \times d}$ with $k \ll d$. To enforce symmetry, we set $W_2 = SW_1$, where $S$ is a diagonal matrix with entries in $\{\pm 1\}$, yielding a low-rank quadratic form with $O(kd)$ parameters.

**4. Object-class / object-identity probes (pointwise).** We first map each embedding to a probability distribution:

$$p = \text{softmax}(W_c x + b), \qquad q = \text{softmax}(W_c y + b),$$

where $W_c$ is trained using multiclass cross-entropy on object-class labels (and similarly $W_N$ for object-identity labels). The pointwise *IsSameObject* score is then defined as the inner product of the two distributions:

$$\textit{IsSameObject}_{\text{class/identity}}(x, y) \; = \; p^\top q \; = \; \sum_{c=1}^{N_c} p(c) \, q(c).$$

### 3.2 *IsSameObject* is best decodable in quadratic form

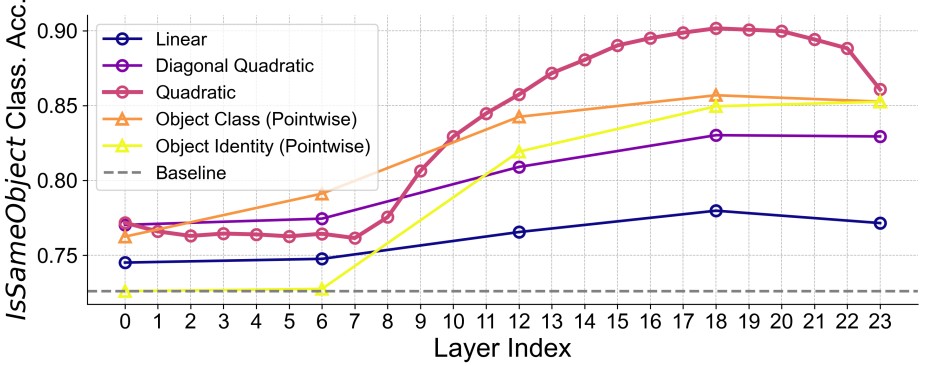

Figure 3: **Quadratic probes excel at decoding the binding signal.** Layer-wise accuracy of the *IsSameObject* probe on DINOv2-Large. The quadratic probe consistently outperforms all other probes from middle layers onward. Results for additional models are shown in Appendix A.3.

We extract DINOv2-Large [13] activations at each layer and train the probes on the ADE20K dataset [56] using cross-entropy loss for all pairwise probes to classify same-object vs. different-object patch pairs (see Figure 2 for *IsSameObject* visualizations). Figure 3 shows probe accuracy across layers. To test our hypotheses about how *IsSameObject* is represented, we compare:

- **Linear < quadratic probes**: (Diagonal) quadratic probes significantly outperform linear ones, suggesting that *IsSameObject* is a quadratic representation, consistent with the quadratic form used by the self-attention mechanism.

- **Quadratic (pairwise) > object-identity probes (pointwise):** Mapping each patch to a discrete object identity and then comparing them pointwise underperforms direct pairwise comparison of embeddings, as the pointwise approach discards information by collapsing continuous representations into discrete classes.

- **Quadratic > object class probes**: The model encodes not only shared object class but also finer-grained identity cues (e.g., distinguishing two identical cars of the same make and model).

- **Full > diagonal quadratic probes**: The *IsSameObject* information is more distributed across dimensions rather than restricted to specific channels.

## 3.3   Object binding emerges broadly across self-supervised ViTs

We extend our analysis beyond DINOv2 to a broader set of pretrained Vision Transformers, including CLIP, MAE, and fully supervised ViTs. To enable direct comparison, we standardize input patch coverage by resizing all inputs so that each model processes the same spatial patch divisions as the DINOv2 family. Under this setup, every probe starts from the same trivial baseline of 72.6% accuracy, which corresponds to always predicting "different", reflecting the class imbalance that most patch pairs do not belong to the same object in the dataset.

Table 1 reports *IsSameObject* decoding accuracy across models. DINO models show the strongest binding signal, with large and giant variants exceeding +16 percentage points over baseline. ImageNet-supervised ViT and CLIP also exhibit clear object-binding ability, though to a lesser degree. In contrast, MAE yields poor object-binding performance, suggesting that binding is an acquired ability under specific pretraining objectives rather than being a universal property of all vision models.

Table 1: **Binding is consistently represented in DINOv2, CLIP and supervised ViT, but less so in MAE.** Probe accuracy on *IsSameObject* across pretrained ViTs. $\Delta$ is reported in percentage points (pp), and the peak layer index is normalized to $[0, 1]$ within each model.

| Model | Highest Accuracy (%) | $\Delta$ over Baseline (pp) | Peak Layer (0–1) |
|---|---|---|---|
| DINOv2-Small | 86.7 | +14.1 | 1.00 |
| DINOv2-Base | 87.5 | +14.9 | 0.82 |
| DINOv2-Large | **90.2** | +17.6 | 0.78 |
| DINOv2-Giant | 88.8 | +16.2 | 0.77 |
| Supervised (ViT-L) | 84.2 | +11.6 | 0.39 |
| CLIP (ViT-L) | 82.9 | +10.3 | 0.65 |
| MAE (ViT-L) | 76.3 | +3.7 | 0.13 |

Our findings thus produce a much wider coverage of ViTs and we provide an understanding of potential reasons why binding emerges:

- **DINO**. The contrastive teacher–student loss enforces consistency across augmented views containing the same objects. This objective encourages the model to learn object-level features that persist under augmented views [9].

- **Supervised ImageNet training**. Although ImageNet labels correspond to the dominant object in each image [57], class-level supervision still provides useful signals for object identity, consistent with the strong performance of our object-class probes.

- **CLIP**. By aligning images with text captions, CLIP effectively assigns each object a symbolic label (e.g., "the red car"), which can act like a pointer that pulls together all patches of that object. This supervision likely encourages patches from the same object to cluster in feature space.

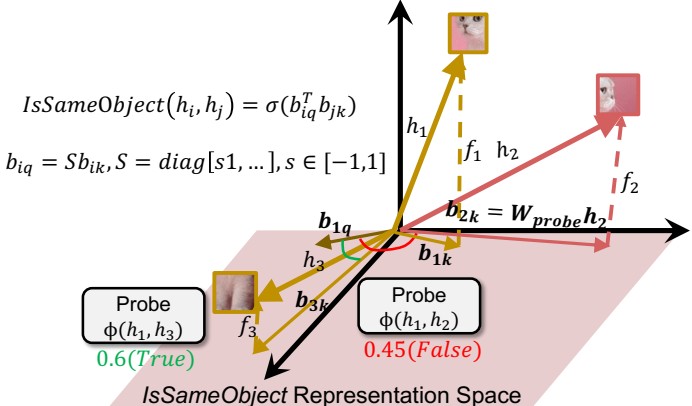

Figure 4: **The geometry of *IsSameObject* representation.** Patch embeddings $h_i$ and $h_j$ are projected onto the *IsSameObject* subspace by $W_1$ (query) and $W_2$ (key), producing binding vectors, whose similarity is computed by a dot product. The asymmetry between query and key spaces (where $W_2$ differs from $W_1$ by sign-flipped rows) mirrors the asymmetric roles of query and key in self-attention.

# 4 Extracting the Binding Subspace of ViT Representations

## 4.1 Decomposing *IsSameObject* from features

Following the linear feature hypothesis [58], and similar to [51], we assume that at layer $\ell$ each token embedding decomposes into a "feature" part and a "binding" part:

$$h^{(\ell)}(x_t) = f^{(\ell)}(x_t, c) \; + \; b^{(\ell)}(x_t),$$

where $f^{(\ell)}(x_t, c) \in \mathbb{R}^d$ encodes all attributes of token $x_t$ (texture, shape, etc.) given context $c = \{x_1, \ldots, x_T\}$, excluding any information about which other tokens it binds with, and $b^{(\ell)}(x_t) \in \mathbb{R}^d$ encodes the binding information that determines which other tokens belong to the same object (i.e., the *IsSameObject* relation).

Consider two identical patches $x_{A_i}$ and $x_{B_i}$ at corresponding positions of identical objects $A$ and $B$ in the same image, and let their residual be $\Delta_{AB_i}$. It may be tempting to cancel the feature term directly. Indeed, without positional encoding (see proof in Appendix A.4.1), we have $f^{(\ell)}(x_{A_i}) = f^{(\ell)}(x_{B_i})$, since for identical tokens the positional encoding is the only signal that can differentiate their cross-token interactions.

We can approximate $f^{(\ell)}(x_{A_i}) \approx f^{(\ell)}(x_{B_i})$, since the two patches are visually identical, appear in nearly the same context, and any positional difference can be offloaded into the binding component. This yields:

$$\Delta_{AB_i} = h(x_{A_i}) - h(x_{B_i}) = \big[f(x_{A_i}) - f(x_{B_i})\big] + \big[b(x_{A_i}) - b(x_{B_i})\big] \approx b(x_{A_i}) - b(x_{B_i}).$$

If $\Delta_{AB_i}$ remains roughly consistent across patch pairs with the same index $i$, then $b(x_{A_i})$ and $b(x_{B_i})$ can form linearly separable clusters, which can thus serve as object identity representations. However, this becomes problematic in natural images, where identical patches are rare.

Instead, we take a *supervised* approach to decoding the binding component. Our quadratic probe serves as a tool for separating binding from feature information within each token (Fig. 4). Conceptually, the quadratic probe can be viewed as projecting an activation $h$ into the *IsSameObject* subspace, yielding $b^{(\ell)}_{query}(x) = h^{(\ell)}(x)^\top W_1$ and $b^{(\ell)}_{key}(x) = h^{(\ell)}(x)^\top W_2$, and then measures the dot-product similarity between two projected vectors. Given that natural image datasets contain numerous objects where $b$ is the primary distinguishing factor, the probe should be optimized to discover a direction that isolates $b$. With this strategy we can separate the binding signal from the rest of the representation.

The observation in [51] that binding vectors remain meaningful under linear combination, and become hard to discriminate when they are close together, is consistent with this interpretation. In later ablation studies, we use our trained quadratic probe via $b^{(\ell)}(x) = h^{(\ell)}(x)^\top W$.

## 4.2 A Toy Experiment: distinguishing identical objects and similar looking objects

To probe the limits of object binding in ViTs, we construct a test image with two identical red cars, a third red car of a different brand, and a red boat. This setup lets us track *IsSameObject* representations across layers by evaluating three distinctions: different object-class but similar appearance, same class with subtle differences, and exact duplicates. As expected, these distinctions become progressively harder. We chose natural objects rather than abstract shapes because both the ViT and our probe are trained on real-world images, which allows us to analyze binding in a nontrivial setting.

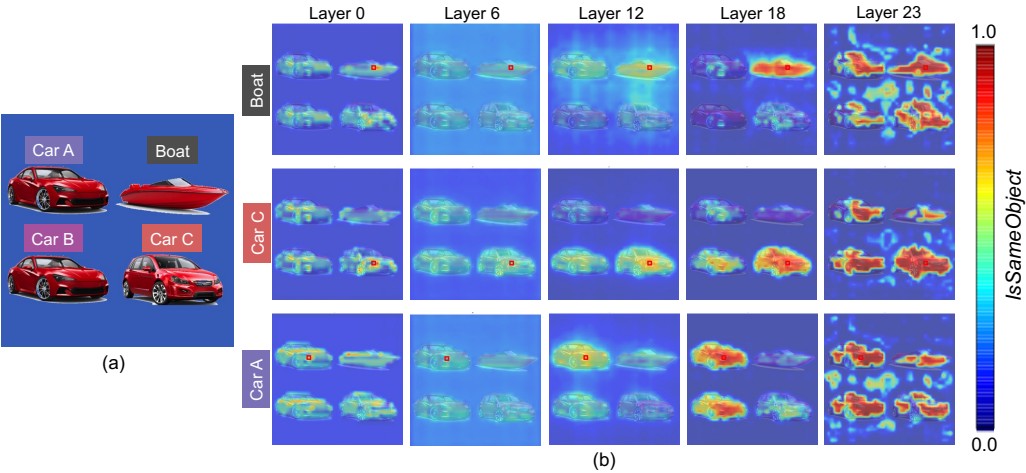

Figure 5: **Layer-wise visualization of *IsSameObject* predictions on the test image.** We used three red cars and one red boat to make binding deliberately difficult. Early layers attend to similar surface features (e.g., the red car or boat hull), mid-layers focus on local objects, and higher layers shift to grouping patches by object class.

To analyze where binding emerges, we plot the *IsSameObject* scores predicted by our trained quadratic probe (Figure 5). We observe that, from early to mid-layers, the model increasingly discerns the local object (the one to which each patch belongs). Surprisingly, from mid-layers to later layers, the model shifts toward class-based grouping, increasingly treating all red cars as the same. Binding emerges in the middle of the network and is then progressively lost towards the top.

**The *IsSameObject* representation is low-dimensional.** We use four identical red-car images and split each one into patches using exactly the same grid alignment. We perform principal component analysis (PCA) on the residuals sets $\{\Delta_{BA}, \Delta_{CA}, \Delta_{DA}\}$, where $\Delta_{BA} = h_{B_i} - h_{A_i} \approx b_{B_i} - b_{A_i}$ and visualize the first three components (see Figure 6). $\Delta_{BA}, \Delta_{CA}, \Delta_{DA}$ fall into three linearly separable clusters in the first three principal component space. The separation of these clusters in a very small number of principal directions demonstrates that *IsSameObject* lies in a low-dimensional subspace: patches from the same object instance map to closely aligned binding vectors, and different instances are linearly separable with large margins.

**Mid-layers capture local objects, and higher layers shift towards grouping patches by object class.** A surprising observation is the sudden increase in the cross-object *IsSameObject* score (Fig.5) in the mid-layers of the DINOV2 model for instances of the same class (Fig. 5). This is consistent with prior work showing that ViTs represent different types of information at different layers [59]. At the same time, token-position decodability drops in deeper layers (see Appendix A.4.3), suggesting that the model is deliberately discarding positional information. Our interpretation is that the network initially relies on positional cues to support binding, since location is necessary to disambiguate tokens that share similar feature content. In later layers, the network removes positional signals once they are no longer useful and repurposes capacity for semantically relevant object structure. Our findings are consistent with experimental evidence from the ventral stream in the brain, showing that while the retinotopic organization of early ventral areas is necessary for perception and binding, global spatial information is instead processed and maintained by the dorsal stream [60–62].

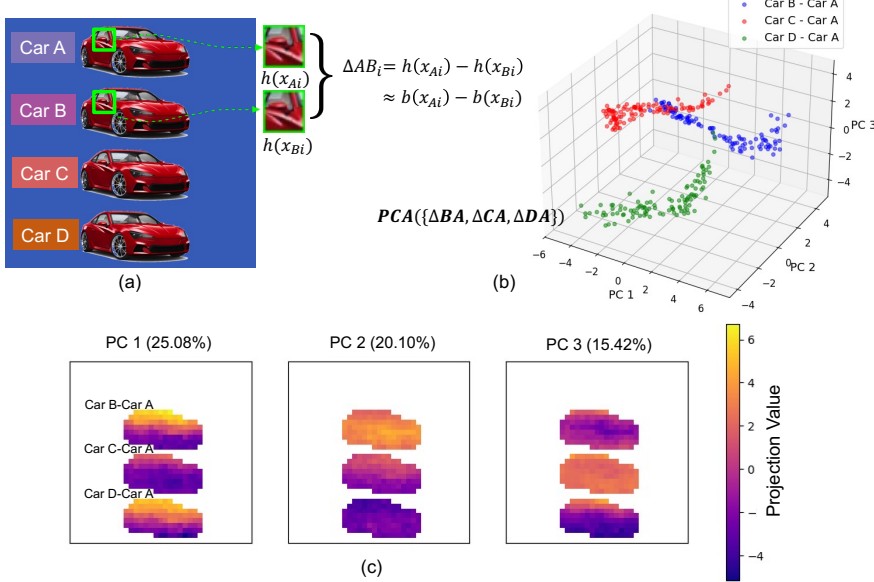

Figure 6: **Identical objects form distinct object-level representations.** The first 3 principal components of the four identical cars, with the three linearly separable clusters denoting $\Delta_{BA}, \Delta_{CA}, \Delta DA$. The percentage in parentheses indicates the variance explained by that principal component.

## 4.3 Attention weights (query-key similarity) correlate with *IsSameObject*

In Section 3.2 we showed that *IsSameObject* is best decoded quadratically. Since self-attention is also a quadratic interaction, binding information in the residual stream at layer $\ell$ can in principle guide how attention is allocated at layer $\ell + 1$, allowing the model to selectively route attention within the same object to build a coherent object-level representation.

To test this, we compute the Pearson correlation between attention weights and the *IsSameObject* scores (see Fig.7 and Appendix A.5). In mid-level layers, we observe a positive but modest correlation, indicating that the model does make use of the *IsSameObject* signal when allocating attention. The modest strength of the effect is expected, because attention serves many roles beyond binding.

## 4.4 Ablation of *IsSameObject* hurts downstream performance and works against the minimization of the pretraining loss

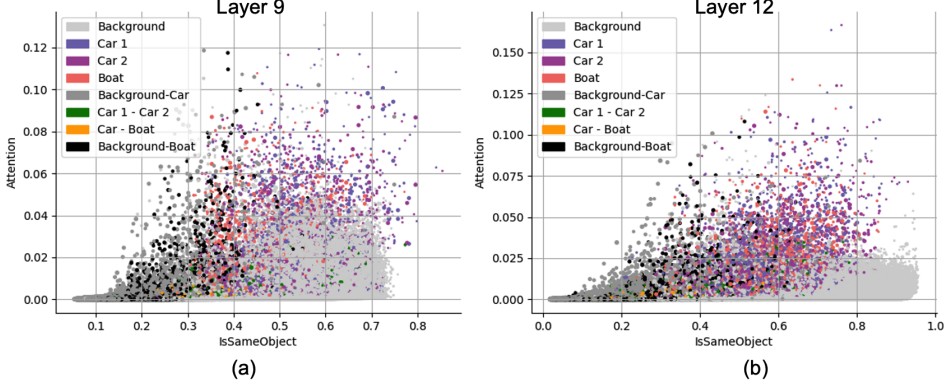

Figure 7: **Attention weights are correlated with *IsSameObject*.** Dot size is proportional to the Euclidean distance between patches. Attention weights correlate with *IsSameObject* in middle layers: (a) Pearson $r = 0.163$, (b) Pearson $r = 0.201$.

Table 2: **Ablations demonstrate the functional role of *IsSameObject*.** Segmentation mIoU, instance mIoU, and DINO loss on layer 18 under uninformed (random shuffle) and informed (ground-truth injection) ablations.

| | **Uninformed** / ratio | | | **Informed** / $\alpha$ | | |
|---|---|---|---|---|---|---|
| | 0 | 0.5 | 1 | 1 | 0.5 | 0 |
| Segmentation mIoU (%) | 44.14 | 41.03 | 39.20 | 44.14 | **44.91** | 43.59 |
| Instance mIoU (%) | 35.14 | 31.39 | 28.19 | 35.14 | 36.37 | **37.02** |
| DINO Loss | 0.6182 | 0.6591 | 0.6749 | 0.6182 | — | — |

We conduct ablation studies and evaluate the impact on downstream segmentation performance and pretraining loss. Instead of directly subtracting the *IsSameObject* representation $b(x_i)$ from $h(x_i)$, we use less aggressive approaches:

- **Uninformed Ablation**: Randomly shuffle $b(x_i)$ across patches in the image at a specified ratio.
- **Informed Ablation (Injection)**: Using ground-truth instance masks, we inject the true *IsSameObject* signal by linearly combining the mean object direction with each patch's binding vector $b_i$: $\tilde{b}_i = (1 - \alpha) \frac{1}{|\mathcal{I}|} \sum_{j \in \mathcal{I}} b_{\text{object},j} + \alpha \, b_{\text{object},i}$.

We evaluate the semantic and instance segmentation performance with retrained segmentation heads on a subset of ADE20K under these variations. We also evaluate the teacher–student self-distillation loss as employed in DINO (see Appendix A.6 for details).

Results show that uninformed ablation, which randomly shuffles the binding vector, reduces segmentation performance, whereas injecting the mean object direction improves accuracy. Ablating *IsSameObject* with random shuffling leads to a noticeable gradual increase in the DINO loss, suggesting that ablation of *IsSameObject* works against this pretraining loss.

## 5 Limitations

We assume the trained probe cleanly splits each patch embedding into "feature" and "binding" components, a simplification that would benefit from further empirical exploration. We do not establish a causal relationship between object binding and downstream task performance, and further analysis is needed to understand how different pretraining objectives induce object binding. Finally, our downstream evaluations focus only on segmentation, leaving open whether these emergent binding signals also benefit other vision tasks such as visual reasoning. More broadly, this paper studies object binding at the patch level; more general forms of binding are not explored and are left for future work.

## 6 Conclusion

In this paper, we show that object binding naturally emerges in large, pretrained vision transformers, especially in DINOv2, and this effect is consistent across multiple models. We also show that it is an acquired rather than innate ability through comparisons across vision models. *IsSameObject*, whether two patches belong to the same object, is reliably decodable and lies in a low-dimensional latent space. Our results emergent object binding arises as a natural solution to self-supervised learning objectives. More broadly, our study bridges what psychologists identify as object binding with emergent behavior in ViTs, challenges the belief that ViTs lack such ability.

Looking ahead, we suggest that addressing binding failures in vision models may not require explicit object-centric modules (e.g., Slot Attention [11]), but could instead be achieved by strengthening the intrinsic object-binding mechanisms of ViTs through tailored training objectives or minimal architectural modifications. Another important direction for future work is to study how bound object representations interact with one another, potentially through low-dimensional "object files" [63]. Together, these efforts will deepen our understanding of how symbolic processing of objects can emerge in connectionist models.

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

# A  Appendix

## A.1  Experimental Setup

**Dataset and Preprocessing.**  Following the DINOv2 standard setup, we use the ADE20K dataset with images resized and cropped to $512 \times 512$ pixels, then padded to $518 \times 518$ pixels. We employ a patch size of $14 \times 14$, resulting in a total of 1,369 patches per image. All computations are performed using float32 precision on a NVIDIA RTX 4090 GPU.

**Training Configuration.**  We use the Adam optimizer with a learning rate of 0.001 and a step learning rate scheduler with step size of 8 epochs and gamma decay factor of 0.2.

All probes are trained for 16 epochs with a batch size of 256 or 128.

## A.2  Probe Details

$$s^{(\ell)} = \mathrm{LayerNorm}\big(h^{(\ell)} + \mathrm{MultiHeadAttention}(h^{(\ell)})\big) \tag{1}$$

$$h^{(\ell+1)} = \mathrm{LayerNorm}\big(s^{(\ell)} + \mathrm{FFN}(s^{(\ell)})\big) \tag{2}$$

Transformers propagate information across layers according to the above equations 12, where $h^{(\ell)}$ is the residual-stream output. We use $h^{(\ell)}$ as the patch embedding for probing at layer $\ell$.

For **pairwise probes**, we apply supervision over the pairwise similarity matrix between all patches in an image. In practice, to reduce computational cost, we randomly sample 64 patches per image in each epoch and apply supervision only to the resulting patch pairs. The probes are trained using binary cross-entropy loss.

For **pointwise probes**, we consider two tasks:

- **Semantic Segmentation**: We use standard cross-entropy loss for pixel-level object class classification.

- **Instance Segmentation**: Following the DETR framework, we employ Hungarian matching for object assignment. The total loss is computed as:

$$\mathcal{L}_{\mathrm{total}} = \lambda_{\mathrm{mask}}\mathcal{L}_{\mathrm{mask}} + \lambda_{\mathrm{dice}}\mathcal{L}_{\mathrm{dice}} \tag{3}$$

  The hyperparameters follow the DETR configuration: `mask_weight` ($\lambda_{\mathrm{mask}}$) = 5.0, `dice_weight` ($\lambda_{\mathrm{dice}}$) = 5.0. Since only 64 patches are sampled per image in each epoch, we use a reduced number of object queries, `num_object_queries` = 10.

### A.2.1  Quadratic probe details

We enforce symmetry in $W$ to reflect the property *IsSameObject(x, y) = IsSameObject(y, x)*. All quadratic probes apply a $1/\sqrt{C}$ normalization. In the full quadratic probe, $\ell_2$ weight decay encourages a low effective rank. The fixed-rank probe instead explicitly upper-bounds $\mathrm{rank}(W)$ through its factorized construction. For analysis, we recover $W1$ and $W2$ from the expression $\sigma\big(x^\top W_1^\top W_2\, y + b\big)$ in Equation 3.1, by computing the singular value decomposition (SVD) of the learned symmetric matrix $W$.

We do not directly parameterize the model using a fixed-rank $W_1$ together with a sign matrix $S$ that defines $W_2$ by flipping rows of $W_1$, as such a discrete sign matrix is difficult to optimize in a fully differentiable manner.

---

**Algorithm 1** Quadratic Probe (full rank)

---

**Require:** $x, y \in \mathbb{R}^C$
 1: Learn $A \in \mathbb{R}^{C \times C}, b \in \mathbb{R}$
 2: $W \leftarrow \frac{1}{2}(A + A^\top)/\sqrt{C}$
 3: $s \leftarrow x^\top W y + b$
 4: **return** $\sigma(s)$

---

**Algorithm 2** Quadratic Probe (with fixed rank $r$)

---

**Require:** $x, y \in \mathbb{R}^C$
1: Learn $U, V \in \mathbb{R}^{r \times C}, b \in \mathbb{R}$
2: $W \leftarrow \frac{1}{2}(U^\top V + V^\top U)/\sqrt{C}$
3: $s \leftarrow x^\top W y + b$
4: **return** $\sigma(s)$

---

## A.3  Probe Performance

### A.3.1  Baselines

We consider several baselines for the *IsSameObject* task.

**Majority baseline.**   As a trivial baseline, we always predict that a patch pair belongs to different objects, reflecting the strong class imbalance in the ADE20K dataset. This baseline achieves an accuracy of 72.6%.

**Distance-based baseline.**   We also include a distance-based baseline that captures dataset statistics. A single scalar threshold is learned on the pairwise patch embedding distance: if the distance is below the threshold, the pair is predicted to belong to the same object; otherwise, it is predicted to be different. This baseline achieves an accuracy of 77.16%.

**Similarity- and attention-based probes.**   We further compare our *IsSameObject* probes against methods motivated by prior work on feature similarity and attention-based object selectivity. For all baselines in this category, only a scalar bias term is trained, while the underlying representations are kept fixed. Specifically, we consider:

- **Cosine similarity probe:** $IsSameObject(x, y) = \sigma\left(-\frac{x^\top y}{\|x\|_2 \, \|y\|_2} + b\right)$;

- **Dot-product probe:** $IsSameObject(x, y) = \sigma(-x^\top y + b)$;

- **Self-attention probe:** $IsSameObject(x, y) = \sigma(-x^\top W_{\text{query}} W_{\text{key}} \, y + b)$, where $W_{\text{query}}$ and $W_{\text{key}}$ are the query and key projection matrices from the self-attention layer at layer $\ell + 1$.

These results (Fig.8) demonstrate that quadratic probes capture *IsSameObject* structure beyond what can be explained by feature similarity or attention alone.

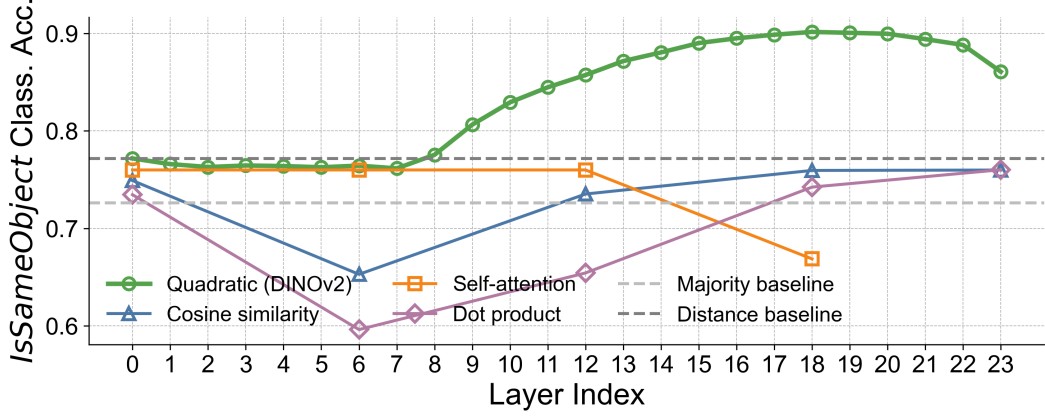

Figure 8: **Layer-wise *IsSameObject* baseline accuracy in DINOv2-Large.**

### A.3.2 Cross-Model Comparison

We train and evaluate quadratic probes on a range of Vision Transformers (ViTs), including the full DINOv2 family (Small, Base, Large, and Giant), as well as CLIP, MAE, and an ImageNet-supervised ViT-Large model. The evaluated backbones are listed in Table 3.

Table 3: Vision Transformer backbones used for evaluation.

| Model family | HuggingFace identifier |
| --- | --- |
| DINOv2-Small | `facebook/dinov2-small` |
| DINOv2-Base | `facebook/dinov2-base` |
| DINOv2-Large | `facebook/dinov2-large` |
| DINOv2-Giant | `facebook/dinov2-giant` |
| CLIP ViT-L/14 | `openai/clip-vit-large-patch14` |
| MAE ViT-L | `facebook/vit-mae-large` |
| ViT-L (IN1K) | `google/vit-large-patch16-224` |

All DINOv2 models use a patch size of $14 \times 14$ pixels and operate on raw inputs of size $518 \times 518$. To enable fair comparison across models with different native input resolutions and patch sizes, we standardize evaluation by matching DINOv2's per-patch spatial coverage. Specifically, inputs to all other models are resized such that their patch grids align with those of DINOv2, resulting in identical spatial patch divisions and, consequently, the same majority baseline accuracy (72.6%).

For example, CLIP ViT-L/14 natively processes $224 \times 224$ images. To match DINOv2's patch grid, the input image is first resized so that its shortest edge is 518 pixels (as in DINOv2), then cropped into multiple $224 \times 224$ regions, each of which is patchified using $14 \times 14$ patches. This procedure ensures consistent spatial correspondence across models despite differences in architecture and pretraining.

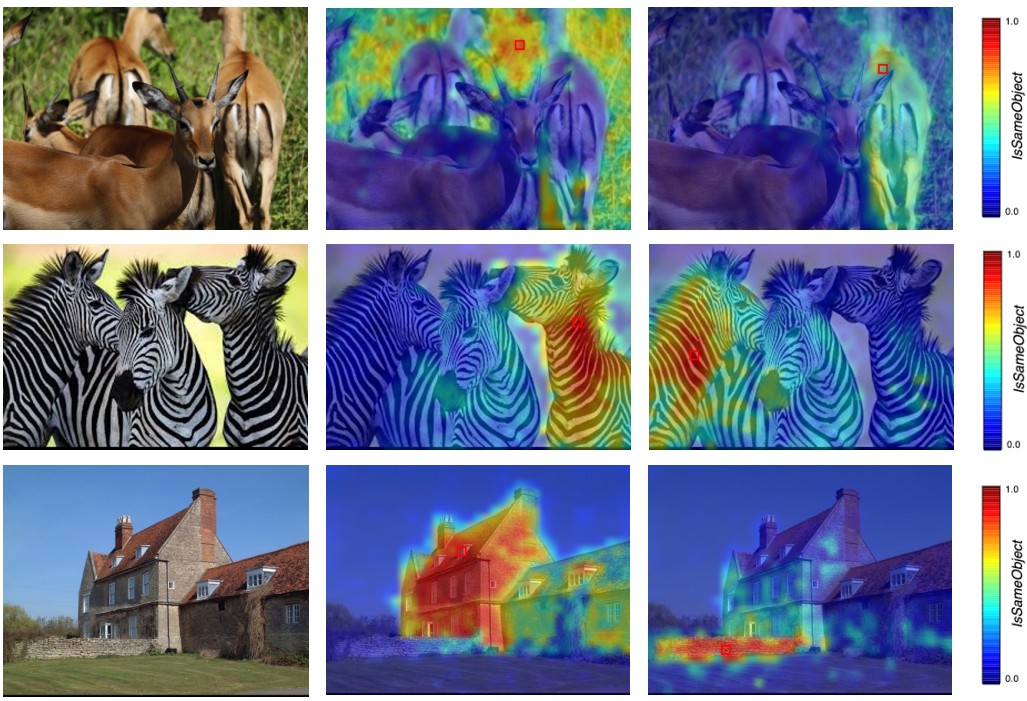

Figure 9: *IsSameObject* **predictions distinguish objects in complex scenes.** Additional visualizations of quadratic probe results for DINOv2-Large at layer 18. The deer image is taken from the DINOv3 paper [53].

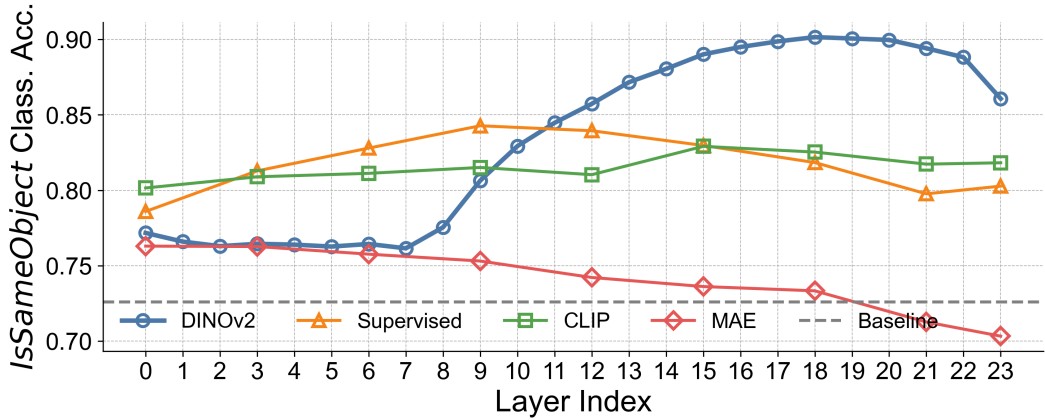

Figure 10: **Layer-wise *IsSameObject* classification accuracy across ViT-Large under different pretraining objectives.** DINOv2 achieves the highest performance, exceeding 90% accuracy in upper layers, while MAE remains close to the baseline, indicating little to no emergent object binding.

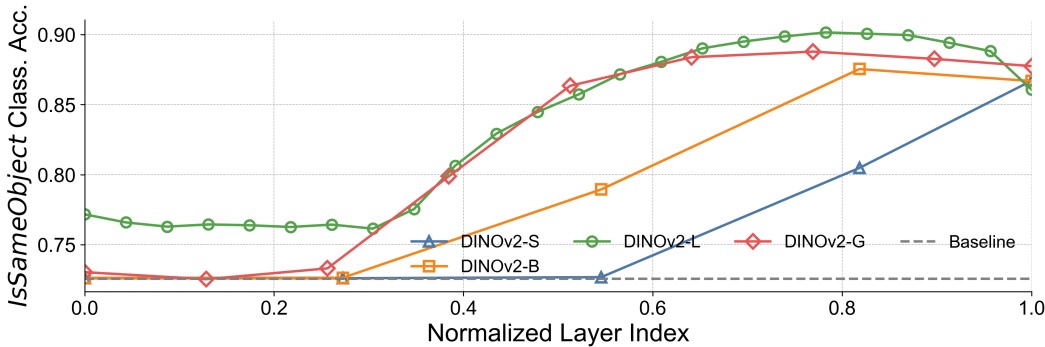

Figure 11: **Layer-wise *IsSameObject* classification accuracy across the DINOv2 family as a function of model size.** DINOv2-Large achieves the strongest overall performance, while peak accuracy is comparable across model scales, reaching approximately 88%.

In Fig. 10, we observe that for ViT-Large models with different pretraining objectives, object-binding performance in DINOv2 begins to emerge in the middle layers and peaks in the later layers, whereas ImageNet-supervised ViT and CLIP remain largely consistent across layers.

In Fig. 11, we observe that within the DINOv2 family, smaller models tend to develop strong object-binding representations at later normalized layers.

## A.4    Positional Information

### A.4.1    Positional Encoding Distinguishes Identical Objects

A single transformer encoder layer can be viewed as comprising two complementary types of computations: token-wise (i.e., position-wise) operations, which act locally on individual tokens and can be executed in parallel to extract features and short-range interactions; and cross-token operations, implemented through scaled-dot-product attention, which enables long-range interactions by integrating contextual information across all tokens.

When inspecting the mathematical formulation, we can also show that for identical tokens (i.e., identical patches), positional encoding is the only information that can guide the cross-token interactions. Here we review the operations for a transformer encoder with a single head from [64] in order.

For simplicity, we also assume that query-key vectors have the same dimension as the model (i.e., $d_k = d$). We use blue color for the token-wise operations and red color for cross-token operations. For a sequence of input tokens $\mathbf{t}_i \in \mathbb{R}^k$ where $k = $ n-channels $\times$ patch-height $\times$ patch width:

pre-processing:
    embedding : $\mathbf{e}_i = \mathbf{t}_i \mathbf{W}_E$
    adding position embedding : $\mathbf{x}_i = \mathbf{e}_i + \mathbf{p}_i$
encoder layer:
    Query-Key-Value : $\mathbf{q}_i = \mathbf{x}_i \mathbf{W}_Q,\ \mathbf{k}_i = \mathbf{x}_i \mathbf{W}_K,\ \mathbf{v}_i = \mathbf{x}_i \mathbf{W}_V$
    self-attention : $\mathbf{U} = \mathrm{softmax}\left(\dfrac{\mathbf{Q}(\mathbf{K})^\top}{\sqrt{d}}\right)\mathbf{V}$
    projection MLP : $\mathbf{y}_i = \mathbf{u}_i \mathbf{W}_O$
    residual connection : $\mathbf{y}_i = \mathbf{x}_i + \mathbf{y}_i$
    normalization : $\mathbf{z}_i = \mathrm{LayerNorm}(\mathbf{y}_i)$
    feed-forward network : $\mathbf{z}_i = \mathrm{ReLU}(\mathbf{z}_i \mathbf{W}_1 + \mathbf{b}_1)\mathbf{W}_2 + \mathbf{b}2$
    residual connection : $\mathbf{z}_i = \mathbf{z}_i + \mathbf{y}_i$

where $\mathbf{W}_E \in \mathbb{R}^{k \times d}$ is the embedding layer, $\mathbf{W}_Q \in \mathbb{R}^{d \times d}$, $\mathbf{W}_K \in \mathbb{R}^{d \times d}$, and $\mathbf{W}^V \in \mathbb{R}^{d \times d}$ are the Query, Key, and Value layers, $\mathbf{W}_O \in \mathbb{R}^{d \times d}$ is the linear projection layer, and $\mathbf{W}_1 \in \mathbb{R}^{d \times m}$, $\mathbf{W}_2 \in \mathbb{R}^{m \times d}$ are the feed-forward weights.

> **Proposition A.1: Positional Encoding Breaks Symmetry**
>
> For a transformer encoder layer, if two input tokens satisfy $\mathbf{t}_i = \mathbf{t}_j$ and $\mathbf{p}_i = \mathbf{p}_j$, then their self-attention outputs are identical, i.e., $\mathbf{u}_i = \mathbf{u}_j$. Therefore, for identical tokens, positional encoding is the only signal that can differentiate their cross-token interactions.

Our goal is to show that for two identical tokens (i.e., two patches with identical features), the transformer has to use the position tagging as cue for binding. Since most operations are token-wise (position agnostic), we only need to show the results for the self-attention operation. We will show that if two input tokens are identical with no positional embedding (or with equal positional embedding), then due to the symmetry of the attention mechanism, their output vectors after self-attention will be identical. Formally, if $\mathbf{t}_i = \mathbf{t}_j \quad i \neq j$ and $\mathbf{p}_i = \mathbf{p}_j$ we want to show that $\mathbf{u}_i = \mathbf{u}_j$.

Assuming $\mathbf{t}_i = \mathbf{t}_j$ and $\mathbf{p}_i = \mathbf{p}_j$:

$$\mathbf{x}_i = \mathbf{t}_i \mathbf{W}_E + \mathbf{p}_i = \mathbf{t}_j \mathbf{W}_E + \mathbf{p}_j = \mathbf{x}_j$$

if $\mathbf{x}_i = \mathbf{x}_j$ then:

$$\mathbf{q}_i = \mathbf{q}_j,\ \mathbf{k}_i = \mathbf{k}_j,\ \mathbf{v}_i = \mathbf{v}_j$$

Thus the attention score computed by $q_i$ and $q_j$ against all keys would be the same:

$$\mathbf{q}_i^\top \mathbf{k}_n = \mathbf{q}_j^\top \mathbf{k}_n \quad \forall n$$

So the attention weights (after softmax) for rows $i$ and $j$ are the same:

$$a_{i,n} = a_{j,n} \quad \forall n \quad \text{where} : \mathbf{a}_i = \mathrm{softmax}\left(\dfrac{\mathbf{q}_i \mathbf{K}^\top}{\sqrt{d_k}}\right)$$

And since the values $\mathbf{V}$ are the same across all inputs for the same $\mathbf{x}_n$, the weighted sum of values will also be identical:

$$\mathbf{u}_i = \sum_{n=1}^{N} a_{i,n} \mathbf{v}_n = \sum_{n=1}^{N} a_{j,n} \mathbf{v}_n = \mathbf{u}_j$$

### A.4.2 Quantifying the Degree of Distinguishing Identical Objects

Here, we use a simplified form of our proposed toy example containing two identical cars and one red boat.

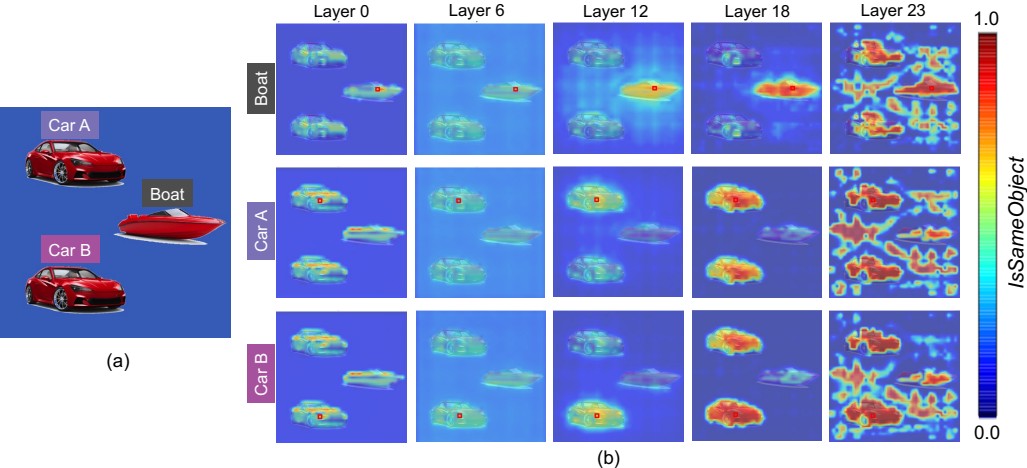

**Figure 12: Layer-wise visualization of *IsSameObject* predictions on the test image with two identical red cars and one red boat.**

We quantify the model's ability to distinguish identical objects by examining the kernel density estimation of *IsSameObject* scores between patch pairs from the same object (Car A, Car B, Boat, or Background) across layers in DINOv2-Large in Figure 12.

Ideally, patch pairs from the same object should achieve *IsSameObject* scores approaching 1.0. In early layers, the distributions cluster around 0.5, indicating the model cannot reliably distinguish same-object from different-object patch pairs. As processing progresses through later layers, these distributions shift toward 1.0. However, some same-object patch pairs continue to score near 0.0 even in deeper layers, representing indistinguishable token pairs.

We also analyzed patch pairs from different objects in Figure 14, where we expect *IsSameObject* scores to approach 0.0. In layers before 12, the distributions correctly cluster near 0.0, showing the model can distinguish different objects. However, as the model learns to group patches within the same object (as shown in the previous analysis), it simultaneously loses its ability to tell the two identical cars apart. This trade-off is visible in the Car1-Car2 distribution, which gradually shifts upward through the layers and develops a strong peak at 1.0 by the final layer.

### A.4.3 Position Information Decay

We hypothesize that the transition from middle layers' capacity to distinguish identical objects to later layers' failure stems from the gradual diffusion of precise positional information into more global, semantically-focused representations. To test this hypothesis, we trained linear probes to decode the (x,y) coordinates of each patch from the model's internal representations (Figure 15). We observe a marked increase in probe RMSE at layer 21, which supports our hypothesis.

### A.5 Attention weights (query-key similarity) vs. *IsSameObject*

We investigate the relationship between attention mechanisms and object identity representations by comparing attention weights with *IsSameObject* scores. Attention weights are computed as:

$$\text{Attention}_{ij} = \text{softmax}\left(\frac{Q_i K_j^T}{\sqrt{d_k}}\right) \tag{4}$$

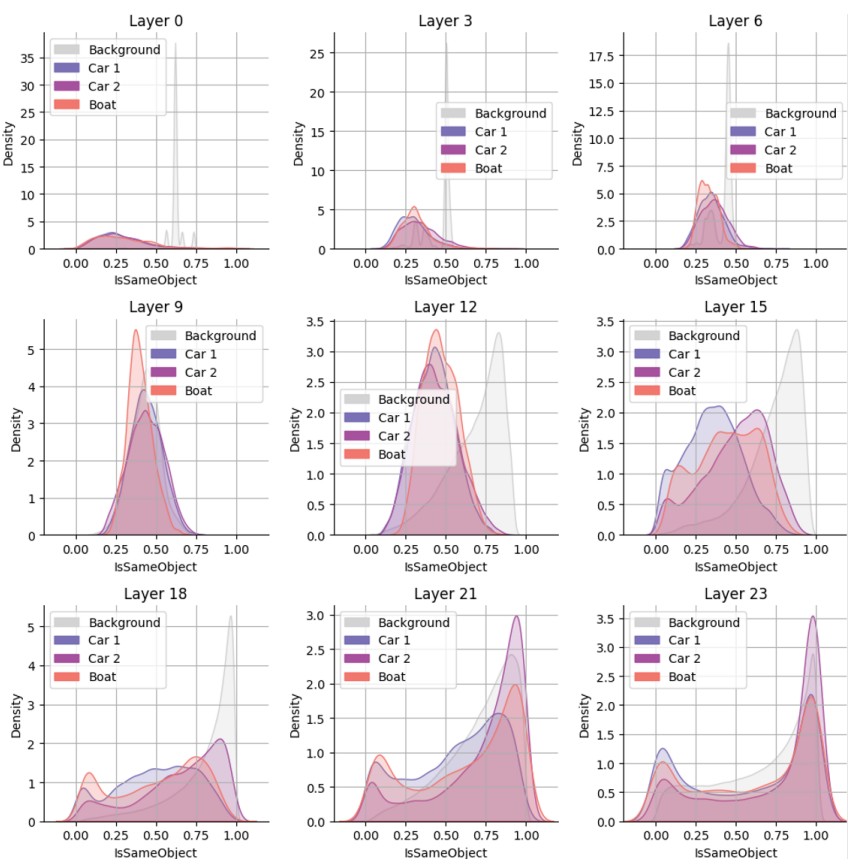

Figure 13: **Binding strengthens with depth for same-object patch pairs.** Kernel density estimation of IsSameObject scores for patch pairs within the same object across different layers.

where $Q_i$ and $K_j$ represent the query and key vectors for patches $i$ and $j$, respectively, and $d_k$ is the key dimension.

We then compute the Pearson correlation between attention weights at layer $\ell + 1$ and *IsSameObject* scores derived from the quadratic probe at layer $\ell$:

$$\rho = \mathrm{corr}\big(\mathrm{Attention}_{ij}^{(\ell+1)}, \mathrm{IsSameObject}_{ij}^{(\ell)}\big). \tag{5}$$

Using the simplified two-car scenario from Figure 12, we examine how attention weights at layer $\ell + 1$ correlate with *IsSameObject* scores at layer $l$. In early layers, we observe minimal correlation between these two measures, which is likely because *IsSameObject* representation has not yet fully developed.

In deeper layers, certain patch pairs receive high attention weights despite having low *IsSameObject* scores (indicating the model believes they belong to different objects). This phenomenon may be explained by background patches being repurposed for internal computational processes, as identified in prior work on DINO register tokens. Future research could further investigate how these specialized background patches contribute to object representation and their role in maintaining distinct "object files".

The Pearson correlations in Fig. 7 are statistically significant (p < 0.001 under permutation test).

## A.6 Implementation of Ablation Studies.

We conduct ablation experiments at layer 18 of DINOv2-Large, where *IsSameObject* representation achieves the best decodability. We apply both uninformed and informed ablation methods as described in Section 4.4.

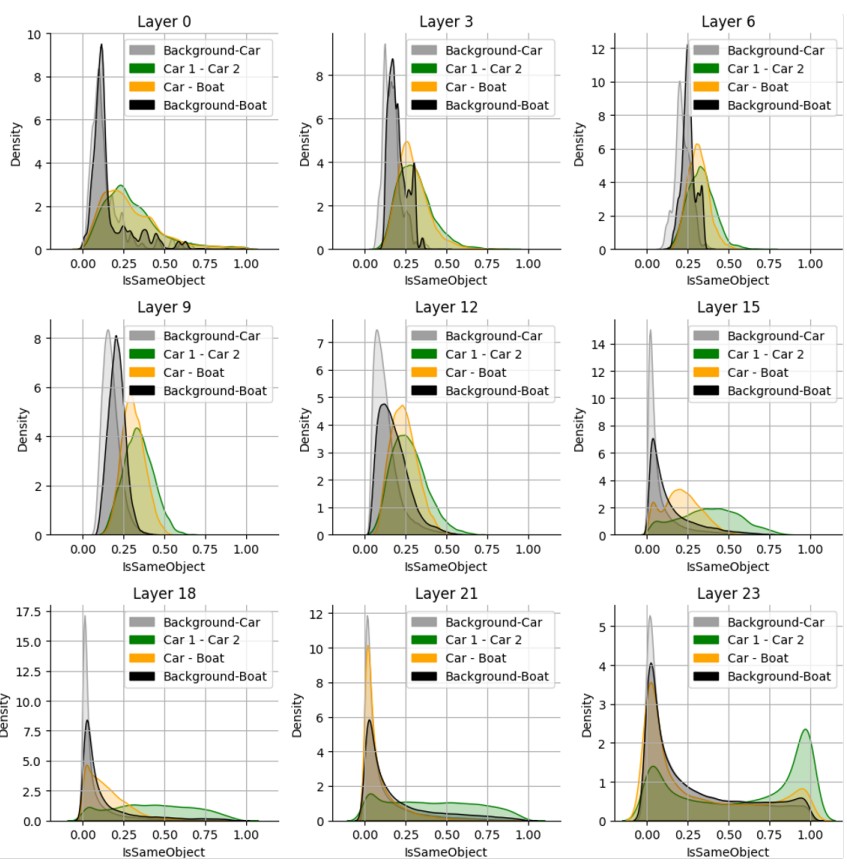

Figure 14: **Identical objects collapse in representation at deeper layers.** Kernel density estimation of *IsSameObject* scores for patch pairs from different objects across layers.

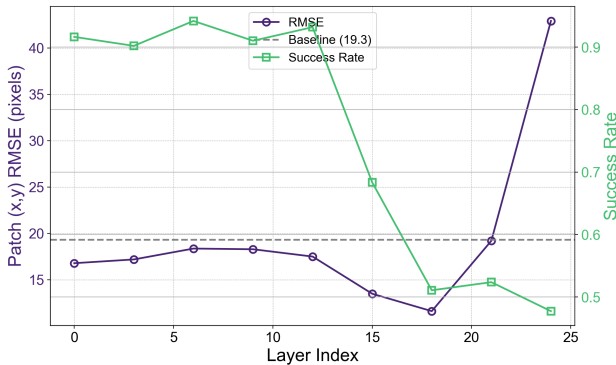

Figure 15: **Positional information decays in later layers.** Layer-wise decoding performance for patch (x,y) coordinates, compared with the success rate of distinguishing patches from Car A and Car B.

**Segmentation Evaluation.** For both semantic and instance segmentation tasks, we retrain linear segmentation heads with ablated representations. The uninformed ablation randomly permutes binding vectors across patches, while the informed ablation injects object-averaged binding vectors using ground-truth masks. These linear heads use identical configurations to the pointwise probes described in Section A.1, which are effectively pointwise probes applied to the final transformer layer.

**DINO Loss Evaluation.** To assess the impact on the pretraining objective, we evaluate DINO loss using the pretrained model as both student and teacher networks. For computational simplicity,

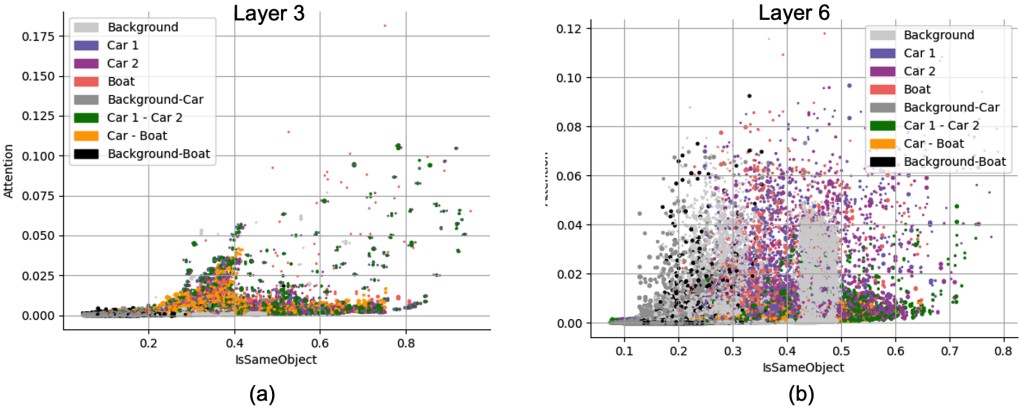

Figure 16: **Attention vs. *IsSameObject* in early layers.** Scatter plots comparing attention weights to *IsSameObject* scores for patch pairs; correlation is still weak, indicating that binding has not yet developed sufficiently to influence attention.

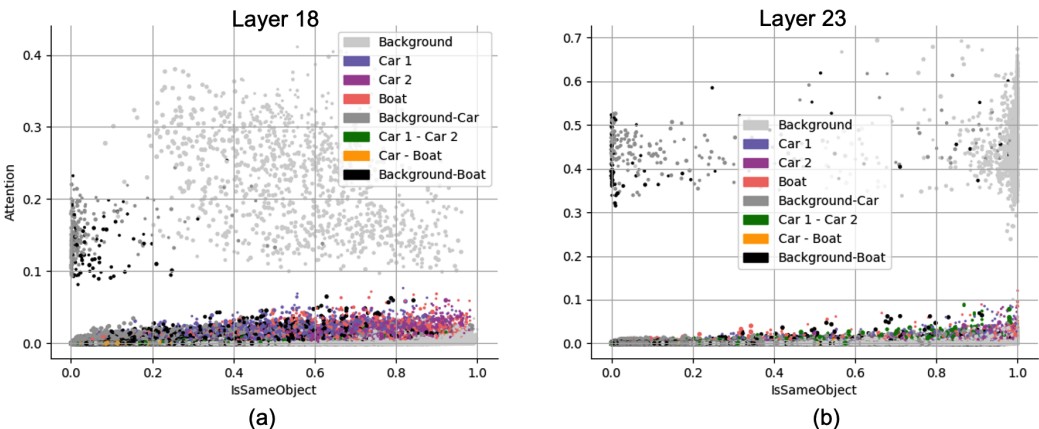

Figure 17: **Attention vs. *IsSameObject* in later layers**. Attention is sometimes allocated to low-*IsSameObject* background tokens, suggesting these tokens might be repurposed for internal computation.

we exclude the iBOT and KoLeo loss components from this analysis. Note that informed ablation cannot be evaluated under DINO loss, as the use of local crops alters the patch divisions, making object-averaged binding vectors undefined for the cropped regions.

