# OpenReview forum: "Does Object Binding Naturally Emerge in Large Pretrained Vision Transformers?"
_NeurIPS.cc/2025/Conference — NeurIPS 2025 spotlight_

### Official Review · Reviewer_4Wjo · 2025-06-28

**Clarity:** 2
**Significance:** 3
**Originality:** 3
**Rating:** 5
**Confidence:** 3

**Summary:**

This work explores whether Vision Transformers (ViTs) develop object binding, which is the ability to group visual features that belong to the same object. The authors propose a pairwise property to test this and show that ViTs do represent such object-based groupings. This representation emerges through layers, aligns with attention patterns, and supports the model’s performance, suggesting that symbolic structure can emerge in connectionist systems without explicit object-centric mechanisms.

**Questions:**

Figure 5 shows that even visually similar objects can be distinguished, which aligns with the linear decomposition hypothesis introduced in Section 3.2. However, in Figure 4, the probe fails to distinguish between different cars. Is this due to the ground truth used during probe training? Specifically, were different instances of the same object class labelled as "same"? If so, have you conducted an experiment using a more fine-grained ground truth, where same-class but different objects are treated as distinct? The results from such an experiment could help reconcile the findings in Figure 4 with the trends shown in Figure 3.

**Ethical Concerns:**

["NO or VERY MINOR ethics concerns only"]

**Final Justification:**

The authors already addressed all my concerns and answered a question. This is a good paper with a comprehensive evaluation. The paper aims to solve a highly relevant problem in both AI and vision science. I would recommend accepting this paper.

**Limitations:**

Yes

**Quality:**

3

**Strengths And Weaknesses:**

$\textbf{Strengths:}$

1. This work contributes to a deeper theoretical understanding of Vision Transformers (ViTs), shedding light on how object-level representations may naturally emerge within their architecture. These insights can inform both our interpretation of ViTs and potential improvements to their design.

2. The results are promising, both quantitatively and qualitatively.

$\textbf{Weaknesses:}$
1. ViT is limited to DINO training. Other ViTs that are trained with different methods (MAE or supervised ViT) are not included. So, the claim of general ViT would be too strong. The tone should be lowered.

2. L157: Object-class and object-identity probes may not work if there are multiple objects from the same class.

3. Figure 4 contrasts with the claim that object binding emerges. For example, Layer 23 images show that four objects are the same (a boat and cars). However, this is acceptable because I would trust the number in Figure 3 more than just one sample.

4. L253: It is unclear how IsSameObject scores are computed. How is correlation between attention weights and IsSameObject scores are computed?


$\textbf{Writing Issues:}$
1. L143 $x_j^l$ should be consistent.

2. The clarity of presentation could be improved.  L153-156 should clarify that W is a learnable parameter, and it should include the details on how to train it.

3. Figure 4 does not have a value for the heatmap.

4. L157: What is 4.5-6? Is it a typo?

---

> ### Author Rebuttal · Authors · 2025-07-31
>
> # Response to reviewer 4Wjo (1/1)
>
> Dear Reviewer 4Wjo,
>
> We thank the reviewer for their careful reading and for recognizing the significance of our research question and robust quantitative results. We appreciate your constructive feedback and are delighted that we were able to run more experiments (across different objectives) and that they substantially improved our paper.
>
> ---
>
> ### **[W1, Q1] Limited verification across pretraining objectives**
> We have now extended our analysis beyond DINOv2 to include CLIP, MAE, and supervised-only trained ViTs (Some limited experiments with CLIP and DINO model sizes were already included in the supplementary: Lines 471-478, Appendix Figure 7). To ensure fair comparisons, we now standardized the effective patch coverage by resizing inputs so that all models process the same spatial patch divisions. This ensures a shared baseline (72.6%) across all models (corresponding to the accuracy of always predicting "different," which accounts for the class imbalance in the dataset).
>
>
> Here are our results:
>
> **Table: Probe accuracy on the *IsSameObject* task across ViTs**
> | **Model**                  | **Highest Probe Accuracy (%)** | **Improvement Over Baseline** | **Layer at Highest Accuracy (Normalized. 0=first layer, 1= last layer)** |
> |:-------------------------- |:-----------------------:|:----------------------------:|:--------------------------:|
> | **DINOv2‑Small**           | 86.7 %                  | + 14.1 pp                    | 1.00                       |
> | **DINOv2‑Base**            | 87.5 %                  | + 14.9 pp                    | 0.82                       |
> | **DINOv2‑Large**           | 90.2 %                  | + 17.6 pp                    | 0.78                       |
> | **DINOv2‑Giant**           | 88.8 %                  | + 16.2 pp                    | 0.77                       |
> | **CLIP (ViT‑Large)**       | 84.2 %                  | + 11.6 pp                    | 0.39                       |
> | **MAE (ViT‑Large)**        | 82.9 %                  | + 10.3 pp                    | 0.65                       |
> | **Supervised (ViT‑Large)** | 76.3 %                  | + 3.7 pp                     | 0.13                       |
>
>
> This now allows us to produce much more interesting insights: CLIP, MAE, and DINO all demonstrate object binding capabilities, supporting our broader claims about *Large Pretrained Vision Transformers* beyond DINOv2 alone. We also show that supervised ImageNet classification yields poor object‑binding performance, indicating that binding is an acquired ability under specific pretraining objectives rather than being a universal property of all vision models.
>
> We offer explanations of why these training regimes work:
> - **DINO**: As noted in Lines 186–189, the DINO loss drives the model toward object-level features by enforcing consistency between augmented views, since each view contains the same objects and the model must align their representations; this objective has been shown to encourage class and object specific features (Caron et al. ICCV 2021), as the reviewer points out.
> - **CLIP**: By aligning images with text captions, CLIP effectively assigns each object a symbolic label (e.g., “the cat”), which can act like a pointer that pulls together all patches of that object. This supervision likely encourages patches from the same object to cluster in feature space. Future work could track object‑binding strength across CLIP training checkpoints to validate this hypothesis.
> - **MAE**: The masked autoencoder objective requires the model to reconstruct a missing patch from its surroundings. When the masked patch sits between multiple objects, correctly predicting its content forces the model to infer which object it belongs to, thus promoting the grouping of patches from the same object.
>
>
> **Our findings thus produce a much wider coverage of ViTs and we provide an understanding of potential reasons for why binding emerges.**
>
> ---
>
> ### **[W2] Probe failure with multiple same‑class objects**
> We agree that object-class and object-identity probes could fail if there are multiple objects from the same class. We’d like to clarify that these probes are used only to test possible hypotheses about how binding works in ViTs: in this case, that the model determines “same‑objectness” by identifying each token’s class or identity and then comparing the token pair.
>
> To evaluate this, we compared probe accuracies across designs: the quadratic probe consistently outperforms both object‑class and object‑identity probes in most layers (Figure 3), indicating that DINOv2‑Large employs a more sophisticated binding strategy than simply classifying tokens by object class or identity and comparing them.
>
>
> ### **[W3] Limited generalizability shown in Figure 4**
> The reviewer questions whether Figure 4 consistently supports our claim of emergent object binding. We’d like to point out that Figure 4 is intended as a stress test, which shows extreme cases with identical objects or same‑class objects in one image to highlight where ViTs may struggle (e.g., distinguishing identical instances or class mates). It is not meant to represent average performance; Figure 3 provides the overall evaluation of object binding across layers and models.
>
> We also note that IsSameObject performance degrades at the final layer (23), likely because DINO’s self‑distillation loss trains only the [CLS] token’s projection‑head output, leaving last‑layer patch‑token activations without direct supervision and prone to drift.
>
> ---
>
> ### **[W4] Ambiguous calculation of IsSameObject scores and correlations**
> IsSameObject scores are the 0–1 outputs of our trained quadratic probe, reflecting the model’s assessed probability that two patches belong to the same object. Attention weights are detailed in Supplementary Lines 530–538. We compute the correlation as the Pearson correlation between all token‑pair IsSameObject scores and their corresponding attention weights.
>
> ---
> ### **[Q] Ground-truth label clarification**
> We’d like to clarify that we always use instance‑level ground truth: different objects, even when they belong to the same class, are always labeled as “different” during probe training and evaluation. Therefore, the confusion seen in Figure 4 is not due to coarse labels but reflects genuine challenges in the model’s binding ability.
>
> Figure 4 evaluates object binding under extreme conditions where multiple identical or same‑class instances in a single image. It shows that, while ViTs do exhibit object binding, it is not flawless: identical objects can still be confused.
>
> ---
> ### **Minor concerns**
> We will correct the listed typos and enhance both our methods' presentation and the rigor of our figures.
>
> We believe that our responses have dramatically strengthened the contributions of our work and are thankful for the reviewer's feedback.
>
> **Best regards,**
> *Authors*

---

> > ### Comment · Reviewer_4Wjo · 2025-08-01
> >
> > Thank you for addressing all my concerns, answering questions, and including new experiments that improve the technical quality of the paper. I appreciate the hard work by the authors, conducting multiple interesting experiments during the rebuttal period.
> >
> > I will increase my score to be accepted, as this is a good paper, and I wish the authors good luck. I hope I can read the polished final version soon.
> >
> > ## [W1, Q1] Limited verification across pretraining objectives
> >
> > Thank you for clarifying this and resolving my concern about the general claim. The experimental results look promising. One thing that really surprises me is the lowest probe accuracy of the supervised ViT (Large).
> >
> > I have one suggestion to improve the clarity of the result. For the baseline, could you please specify what kind of baseline strategy you use? I understand the baseline as using random guessing, and its accuracy is 72% because of imbalanced classes. Am I understanding correctly?
> >
> > Lastly, if you were to speculate, what would you think the reasons are that make supervised ViT the worse, regardless of the rich label signals and high performance in image classification? Do you think this is because of shortcuts learned by supervised ViT?  Moreover, DINOv2 giant also has lower probe accuracy than DINOv2 large, which should be the opposite trend to downstream performance. I just want to clarify that these results are not a limitation, but they are interesting phenomena, so there is no need to conduct experiments during the discussion period or even in the main paper. You could consider it as future work if you think it is worth exploring.
> >
> > ## [W2] Probe failure with multiple same‑class objects
> >
> > Thank you for answering this.
> >
> > ## [W3] Limited generalizability shown in Figure 4
> > This answer is clear. I acknowledge it.
> >
> > ## [W4] Ambiguous calculation of IsSameObject scores and correlations
> > The answer is clear to me. I suggest that you might include the mathematical formulation of how the correlation between attention weights and IsSameObject scores is computed in the final version (at least in the appendix). It's just for self-contained purposes, and you may not need to do it.
> >
> >
> > ## [Q] Ground-truth label clarification
> >
> > This is an interesting point. I am excited to see future work studying this problem, especially how to improve the emergence of object binding. Do you think embodiment signals can help with this? Humans may bind objects based on how they interact with them. It is interesting to see if the vision models could gain the same ability without direct supervision.

---

> > > ### Author Response · Authors · 2025-08-01
> > >
> > > Thank you for your prompt feedback and for recognizing the value of the new experiments and clarifications. We’re glad they strengthened the paper. Below are our responses to your comments.
> > >
> > > ---
> > >
> > > ### **[W1] Limited verification across pretraining objectives**
> > > - **Baseline strategy**: We always predict “different” (IsSameObject = 0) for every token pair. On the ADE20K test split we use, 72.6% of token pairs come from different objects and 27.4% from the same object, so this constant prediction yields 72.6% accuracy. It is not random guessing but a fixed, trivial baseline (“always-predict-different”). We will clarify this point in the final version.
> > > - **What explains supervised ViT’s poorer performance?**
> > >     - The supervised-ViT was pretrained via supervised image classification on ImageNet, which contains only ≈20% multi-object images and ≈80% with a single dominant object (Shankar et al. 2024; Anzaku et al. 2024). Therefore, the training signal provides little incentive to handle multiple objects or to learn the fine-grained, multi-patch relational structure that object binding requires.
> > >     - Several studies show that supervised image classification encourages **shortcut** learning: the model often relies on spurious cues or irrelevant regions (e.g., texture or background context) that suffice for the class label. For example, when telling huskies and wolves apart, classifiers trained on data where wolves appear more often in snowy scenes can rely on the presence of snow in the background instead of the animals’ actual features (Ribeiro et al. 2016). In this sense, future work on improving binding may also promise to improve real world performance of DL models.
> > >     - We also plan to test whether this finding holds consistently across model scales.
> > > - **DINOv2-Giant underperforms DINOv2-Large in object binding**: The DINOv2 paper reports that Dino-Giant outperforms Dino-Large on downstream tasks. Object binding is not a direct proxy for downstream performance and raises interesting questions on whether some other property of the model compensates, or if other binding measures are more predictive of downstream performances (e.g., the ability to distinguish objects from the same class).
> > >
> > > ### **[Q] Ground-truth label clarification**
> > > *How can we improve object binding?* As the reviewer points out, one promising direction is **embodiment**: humans use correlated action–perception loops—manipulating objects, observing the consequences, and integrating proprioceptive and temporal feedback—to infer which parts move or behave coherently. In the end, both animals and models deployed on real tasks rely on affordances (Gibson 1979)—the actions the world makes possible—so making them explicit should better align representations with use cases. Even without active human interaction, object persistence (i.e., temporal coherence across frames) provides strong cues. All of these are promising directions for future work.
> > >
> > > ### [W2, W3, W4]
> > > Thank you for acknowledging our responses. To improve clarity, we will include the correlation calculations in the method.
> > >
> > > ---
> > > Thank you again for your insightful comments, which point to promising directions for future work. We’ll work hard to deliver a strong final version that we hope will be valuable to the community.

---

### Official Review · Reviewer_gvWE · 2025-07-03

**Clarity:** 3
**Significance:** 3
**Originality:** 2
**Rating:** 4
**Confidence:** 4

**Summary:**

his paper examines whether object binding emerges naturally in pre-trained ViTs, introducing the concept IsSameObject to indicate whether two patches belong to the same object. Using a similarity probe, the authors show that this signal is decodable, lies in a low-dimensional space, correlates with attention, and affects downstream performance. The study suggests that object binding can arise in ViTs without explicit modules.

**Questions:**

1. Generalization to Other ViTs: How does the IsSameObject signal behave in ViTs trained under different paradigms (e.g., MAE, CLIP)?

2. Low-Dimensional Subspace: Why is it meaningful that IsSameObject lies in a low-dimensional subspace? Does this suggest compression, generalization, or modularity—and how could it be exploited in practice?

3. Visualization and Interpretability: In Figure 4, the visualization of layer 23 does not support the conclusion that higher layers shift to grouping patches by object class. For example, feature from the boat also largely attends to the cars. Can the author provide more convincing visulizations?

**Ethical Concerns:**

["NO or VERY MINOR ethics concerns only"]

**Final Justification:**

The rebuttal convincingly addressed major concerns by broadening experiments beyond DINOv2 to CLIP, MAE, and supervised ViTs with standardized baselines, adding cross-layer binding analysis, clarifying claims about ViT limitations with stronger neuroscientific context, and providing further attention–binding correlation evidence. Methodological details (baselines, projection methods, ablations) were clarified, improving reproducibility. Core strengths—novel framing, rigorous experiments, and relevance to both CV and neuroscience—remain strong. Minor concerns on efficiency profiling and modest ablation effect sizes are acknowledged but carry low weight. Overall, the work is technically sound, timely, and impactful, warranting acceptance.

**Limitations:**

Yes.

**Paper Formatting Concerns:**

No.

**Quality:**

2

**Strengths And Weaknesses:**

Strengths:

Novel Hypothesis and Probe Design: The introduction of IsSameObject as a testable signal is both conceptually interesting and grounded in the architecture of ViTs.

Empirical Rigor: Strong quantitative results showing high probe accuracy, supported by ablation studies and correlation with attention weights.

Theoretical Insight: Bridges cognitive science (object binding) with deep learning, providing an interpretable lens through which to view ViT behavior.

Practical Implications: Challenges the necessity of explicit binding modules and suggests more implicit, architecture-aligned approaches.

Weaknesses:

Limited Scope of Models: While DINOv2 is emphasized, it’s unclear how generalizable the findings are across a broader range of architectures or training paradigms.

Limited Visual Support: The existing visualizations may not fully substantiate the claim that mid-layers capture local objects, and higher layers shift towards grouping patches by object class.

Unclear Utility of Low-Dimensional Subspace: While IsSameObject is shown to lie in a low-dimensional subspace, the practical or theoretical implications of this finding are not well articulated. It's unclear why this matters or how it can be used.

---

> ### Author Rebuttal · Authors · 2025-07-31
>
> # Response to reviewer gvWE (1/1)
>
> Dear Reviewer gvWE,
>
> We thank the reviewer for their careful reading and for recognizing our novel *IsSameObject* hypothesis, probe design, and robust quantitative results. We appreciate your feedback and are delighted that we were able to run experiments across different objectives and that they substantially improved our paper.
>
> ---
>
> ### **[W1, Q1] Limited verification across pretraining objectives**
> We have now extended our analysis beyond DINOv2 to include CLIP, MAE, and supervised-only trained ViTs (Some limited experiments with CLIP and DINO model sizes were already included in the supplementary: Lines 471-478, Appendix Figure 7). To ensure fair comparisons, we now standardized the effective patch coverage by resizing inputs so that all models process the same spatial patch divisions. This ensures a shared baseline (72.6%) across all models (corresponding to the accuracy of always predicting "different," which accounts for the class imbalance in the dataset).
>
> Here are our results:
>
> **Table: Probe accuracy on the *IsSameObject* task across ViTs**
> | **Model**                  | **Highest Probe Accuracy (%)** | **Improvement Over Baseline** | **Layer at Highest Accuracy (Normalized. 0=first layer, 1= last layer)** |
> |:-------------------------- |:-----------------------:|:----------------------------:|:--------------------------:|
> | **DINOv2‑Small**           | 86.7 %                  | + 14.1 pp                    | 1.00                       |
> | **DINOv2‑Base**            | 87.5 %                  | + 14.9 pp                    | 0.82                       |
> | **DINOv2‑Large**           | 90.2 %                  | + 17.6 pp                    | 0.78                       |
> | **DINOv2‑Giant**           | 88.8 %                  | + 16.2 pp                    | 0.77                       |
> | **CLIP (ViT‑Large)**       | 84.2 %                  | + 11.6 pp                    | 0.39                       |
> | **MAE (ViT‑Large)**        | 82.9 %                  | + 10.3 pp                    | 0.65                       |
> | **Supervised (ViT‑Large)** | 76.3 %                  | + 3.7 pp                     | 0.13                       |
>
>
> This now allows us to produce much more interesting insights: CLIP, MAE, and DINO all demonstrate object binding capabilities, supporting our broader claims about *Large Pretrained Vision Transformers* beyond DINOv2 alone. We also show that supervised ImageNet classification yields poor object‑binding performance, indicating that binding is an acquired ability under specific pretraining objectives rather than being a universal property of all vision models.
>
>
> We offer explanations of why these training regimes work:
> - **DINO**: As noted in Lines 186–189, the DINO loss drives the model toward object-level features by enforcing consistency between augmented views, since each view contains the same objects and the model must align their representations; this objective has been shown to encourage class and object specific features (Caron et al. ICCV 2021), as the reviewer points out.
> - **CLIP**: By aligning images with text captions, CLIP effectively assigns each object a symbolic label (e.g., “the cat”), which can act like a pointer that pulls together all patches of that object. This supervision likely encourages patches from the same object to cluster in feature space. Future work could track object‑binding strength across CLIP training checkpoints to validate this hypothesis.
> - **MAE**: The masked autoencoder objective requires the model to reconstruct a missing patch from its surroundings. When the masked patch sits between multiple objects, correctly predicting its content forces the model to infer which object it belongs to, thus promoting the grouping of patches from the same object.
>
>
> **Our findings thus produce a much wider coverage of ViTs and we provide an understanding of potential reasons for why binding emerges.**
>
>
> ---
> ### **[W2, Q3]  Limited visual support**
> Although we cannot include visualizations at this stage, we have validated our claim that mid‑layers capture local object structure and higher layers emphasize object‑class grouping by training new probes. The *IsSameObjectClass* probe’s accuracy increases in deeper layers as *IsSameObject* accuracy declines, directly supporting this shift. *IsSameObjectClass* accuracy drops at the final layer (23), likely because DINO’s self‑distillation loss targets only the [CLS] token’s projection‑head output—meaning the last‑layer patch‑token activations receive no direct training signal and can drift away from class‑level structure.
>
> We also introduced *IsSameGreyscale* probe for low‑level perceptual features; its accuracy decreases rapidly with depth, suggesting a rapid shift from low-level perceptual features to high-level semantics.
>
>
> **Table: Mean probe accuracy (%) across layers for *IsSameObject*, *IsSameObjectClass*, and *IsSameGreyscale*.**
> | Layer | IsSameObject | IsSameObjectClass | IsSameGreyscale |
> |:-----:|-------------:|------------------:|------------:|
> | 0 | 40.4% | 40.2% | 91.2% |
> | 6 | 48.2% | 50.1% | 69.4% |
> | 12 | 75.2% | 74.3% | 49.7% |
> | 18 | 70.3% | 85.5% | 25.8% |
> | 23 | 50.1% | 53.1% | 20.9% |
>
> ---
>
> ### **[W3, Q2] Implications of low-dimensional *IsSameObject* space**
>
> A low‑dimensional IsSameObject subspace indicates that the binding information is concentrated in just a few directions. This makes learning of those directions which matters for downstream tasks and attention far easier. In this low-dimensional space, we also find that there are large margins between the clusters (Lines 231-237, Figure 5), which makes them easy to separate and guarantees generalization. This type of representation not only supports strong generalization (because large margins between clusters imply robustness to new inputs), but also enables efficient compression: by projecting full token embeddings onto the top binding dimensions, we can distill the binding signal into a much smaller code.
>
> We believe that our responses have dramatically strengthened the contributions of our work and are thankful for the reviewer's feedback.
>
> **Best regards,**
> *Authors*

---

> > ### Comment · Reviewer_gvWE · 2025-08-05
> >
> > Thanks for the detailed rebuttal. My concerns have been addressed, and I have no more questions. I will raise my rating to 4.

---

> > > ### Author Response · Authors · 2025-08-06
> > > **Thank you!**
> > >
> > > We're glad we could address your concerns. Thank you for your constructive suggestions and positive assessment of our paper!

---

### Official Review · Reviewer_x2fh · 2025-07-05

**Clarity:** 2
**Significance:** 3
**Originality:** 3
**Rating:** 5
**Confidence:** 3

**Summary:**

This papers tackles the question of whether object binding emerges within large pre-trained vision transformers (focusing on DINO). The binding problem refers to how the brain is able to group disparate visual features with in a parallel distributed system to a single coherent whole to represent an object. This paper extends this problem to computer vision by trying to understand whether vision transformers exhibit any object binding mechanisms. The paper proposes training probes that can detect whether two patches belong to the same object and how that such probes can be successfully trained on patch embeddings. Furthermore, the paper presents results showing that ablating the ability of the model to detect isSameObject can be detrimental to performance.

**Questions:**

- Do the results apply to ViTs in general? If so, can you provide evidence that similar binding strategies hold in networks trained with other supervision?
- Can you please comment on the choice of only studying binding within each layer while it was intended to measure how the brain binds information across different brain areas?
- The paper states that there is a current view held by cognitive scientists that VITs lack object binding. Could you please provide evidence that this is a commonly held view in the cognitive science community?
- Could you comment on the concerns raised about the experimental results and the conclusions in Sec 4.3 and 4.4.
- What is the baseline referred to in Fig 3? Is it just performance of the linear probe at the first layer?
- Could you please explain how the binding vectors are extracted from the token embeddings? It seems clear that difference of embeddings (h_b - h_a) equates to difference of binding vectors (b_b - b_a). However, it's unclear how to decompose h_a into f_a and b_a which seem necessary to conduct the experiments in Sec 3.3.

**Ethical Concerns:**

["NO or VERY MINOR ethics concerns only"]

**Final Justification:**

As noted in my initial review, the submission was already strong. There were some limitations regarding framing and experimental validation that were addressed in the rebuttal. I believe the resulting paper will be much stronger and will raise my rating to Accept.

**Limitations:**

While there is a limitations section that highlights two limitations of the work, it does not seem adequate to me. As noted in the weaknesses, the object binding problem is complex and mapping it to ViTs has to come with many assumptions that limit the claims that can be made about the work.

**Paper Formatting Concerns:**

None.

**Quality:**

3

**Strengths And Weaknesses:**

**Strengths:**
- The introduction and related work are very well written and very nicely set up the problem and contextualize it.
- The binding problem is an interesting problem within connectionist systems, and I really appreciated the effort put to explain its significance as well as what it entails. As well as the delineation between efforts that try to solve it architecturally (eg, Slot Attention) as opposed to the question posed by this paper of trying to find its mechanisms within network.
- I appreciate the care taken experimentally to differentiate between different hypothesis in Sec 3.1.

----
**Weaknesses**: I've tried to order the weaknesses in terms of importance/significance.
- The title and framing of the paper does not match the experimental setup. The paper sets itself as measuring whether object binding emerges within "pretrained vision transformers." This can be seen in both the title as well as the framing in the introduction (L50-71). However, the paper primarily focuses on DINO (with one experiment also reporting results on CLIP). This is a major confounder as DINO's loss results in class and object specific features; this was indeed one of the main claims in Caron et al (ICCV 2021). Without showing results that indicate that transformers learn this with different training regiments.
- Object binding is concerned with how to relate perceptual representations that "depend on distributed neural codes for relaying the parts and properties of objects." [Treisman, 1966]. On the other hand, this paper focuses on binding within each layer with little discussion of whether there are any binding mechanisms across layers (or a way to preserve the isSameObject across layers). This is important as some work has shown that ViTs represent different types of information at different layers (eg Amir et al. ECCVW 2022) which can be seen qualitatively in Figure 4b.
- The abstract states that the findings "challenge the view that ViTs lack object binding" this is followed by similar claims in L51-53 with references 11, 12, 7. However, the references don't seem to provide evidence for this. Ref11 is a paper that proposes a clustering mechanism to improve performance on human pose estimation. Ref12 on the other hand doesn't make any mention of ViTs or self-attention, and the critique for computer vision approaches seems to focus on feedforward convolutional networks. It's unclear from the evidence provided that there's a strong view within the cognitive science community that ViTs lack this ability.
- Some of the claims in Section 4 don't seem to be supported by the results:
    - In Sec 4.3, the paper claims that attention weight correlate with isSameObject. The correlation plot shown in Figure 6 shows a strong spread even if it's exhibits a positive trend (as indicated by the r values of 0.163/0.201). Furthermore, isSameObject is a proxy estimate for whether two patches belong to the actual object. It would make sense to show that the attention is stronger between patches of the same object as opposed to patches belonging to different object of the same class (or different objects)?
    - In Sec 4.4, the paper states that ablating isSameObject hurts downstream performance. First, it's unclear how this is ablated as Section 3.2 shows how the difference of features correlates to the difference of the binding parts. However, it doesn't explain how you can obtain the binding parts from the features. Second, the trends shown in Table 1 exhibit performance changes that (1) don't show monotonic improvement/regression as shown by segmentation accuracy for informed ablation and (2) seem like fairly small differences in performance without being supported by some baseline to calibrate expected model changes.
- There are several experimental details that aren't clear from the paper. See questions.

----
**Minor concerns/comments**
- There seems to be some latex/formatting issues in L141 (equations 1/2) and L157 (4.5-6).
- I found the latter parts for section 3 harder to follow as they explain different experiments before showing the results sequentially in section 4. I would suggest that you consider interleaving the experimental results with the method which could improve clarity and readability.

---

> ### Author Rebuttal · Authors · 2025-07-31
>
> # Response to reviewer x2fh (1/1)
>
> Dear Reviewer x2fh,
>
> We thank the reviewer for your careful reading, constructive feedback, and positive comments on our contextualization and emphasis of the research problem’s significance. We are delighted that we were able to run the requested experiments (across different objectives) and that they substantially improved our paper.
>
> Below, we address your comments point by point.
>
> ---
>
> ### **[W1, Q1] Limited verification across pretraining objectives**
> We have now extended our analysis beyond DINOv2 to include CLIP, MAE, and supervised-only trained ViTs (Some limited experiments with CLIP and DINO model sizes were already included in the supplementary: Lines 471-478, Appendix Figure 7). To ensure fair comparisons, we now standardized the effective patch coverage by resizing inputs so that all models process the same spatial patch divisions. This ensures a shared baseline (72.6%) across all models (corresponding to the accuracy of always predicting "different," which accounts for the class imbalance in the dataset).
>
> Here are our results:
>
> **Table: Probe accuracy on the *IsSameObject* task across ViTs**
> | **Model**                  | **Highest Probe Accuracy (%)** | **Improvement Over Baseline** | **Layer at Highest Accuracy (Normalized. 0=first layer, 1= last layer)** |
> |:-------------------------- |:-----------------------:|:----------------------------:|:--------------------------:|
> | **DINOv2‑Small**           | 86.7 %                  | + 14.1 pp                    | 1.00                       |
> | **DINOv2‑Base**            | 87.5 %                  | + 14.9 pp                    | 0.82                       |
> | **DINOv2‑Large**           | 90.2 %                  | + 17.6 pp                    | 0.78                       |
> | **DINOv2‑Giant**           | 88.8 %                  | + 16.2 pp                    | 0.77                       |
> | **CLIP (ViT‑Large)**       | 84.2 %                  | + 11.6 pp                    | 0.39                       |
> | **MAE (ViT‑Large)**        | 82.9 %                  | + 10.3 pp                    | 0.65                       |
> | **Supervised (ViT‑Large)** | 76.3 %                  | + 3.7 pp                     | 0.13                       |
>
> This now allows us to produce much more interesting insights: CLIP, MAE, and DINO all demonstrate object binding capabilities, supporting our broader claims about *Large Pretrained Vision Transformers* beyond DINOv2 alone. We also show that supervised ImageNet classification yields poor object‑binding performance, indicating that binding is an acquired ability under specific pretraining objectives rather than being a universal property of all vision models.
>
> We offer explanations of why these training regimes work:
> - **DINO**: As noted in Lines 186–189, the DINO loss drives the model toward object-level features by enforcing consistency between augmented views, since each view contains the same objects and the model must align their representations; this objective has been shown to encourage class and object specific features (Caron et al. ICCV 2021), as the reviewer points out.
> - **CLIP**: By aligning images with text captions, CLIP effectively assigns each object a symbolic label (e.g., “the cat”), which can act like a pointer that pulls together all patches of that object. This supervision likely encourages patches from the same object to cluster in feature space. Future work could track object‑binding strength across CLIP training checkpoints to validate this hypothesis.
> - **MAE**: The masked autoencoder objective requires the model to reconstruct a missing patch from its surroundings. When the masked patch sits between multiple objects, correctly predicting its content forces the model to infer which object it belongs to, thus promoting the grouping of patches from the same object.
>
> **Our findings thus produce a much wider coverage of ViTs and we provide an understanding of potential reasons for why binding emerges.**
>
> ---
> ### **[W2, Q2] Cross-layer binding**
>
> We thank the reviewer for highlighting cross‑layer interactions as a novel and necessary perspective on studying binding in ViTs; since brains bind across layers, we ask whether ViTs do too.
>
> Cross-layer binding may actively happen under these conditions:
> - **Layer-specific information processing**: ViTs represent different types of information at different layers. Although the self-attention implements a "full range" interaction (all tokens to all tokens interaction), the model learns to limit the "interaction" between the correct tokens and integrate features of the object they belong to as we go deeper.
> - **Information retrieval from earlier layers**: ViTs may access information from earlier layers when current layers lack necessary binding cues. As shown in Appendix Figure 11 and in Amir et al. ECCVW 2022, positional information is gradually removed as the network deepens, and other low-level features may also be discarded. This loss may lead the model to retrieve binding cues from earlier layers instead of keeping them redundantly; evaluating this idea needs further work.
>
> Cross-layer interactions in ViTs can occur via the residual skip connections, so we also train quadratic probes between non-adjacent layers. Specifically, we compute $\phi(x,y) = x^TW_1^T W_2 y$, where $W_{1}$ and $W_{2}$ are the learned projection matrices from layer 15 and layer 18, respectively. The probe accuracies are:
>
> - Layer 15: 89.0%
> - Layer 18: 90.1%
> - Cross (15→18): 83.3%
>
> The cross-layer probe shows decent cross-layer object binding, and a certain degree of object binding across the system.
>
> ---
> ### **[W3, Q3] Consensus on ViT object‑binding limitations**
>
> We agree that our claim was not well articulated and could lead to misunderstanding. We should clarify that the cognitive- and neuro-scientists believe that binding requires a set of interacting mechanisms such as memory and recurrent processing trained on complex behavior (please see Roelfsema 2023, Solving the binding problem; Peters et al. 2021, Capturing the objects of vision with neural networks; Salehi et al. 2024, Modeling Attention and Binding in the Brain through Bidirectional Recurrent Gating). Therefore, our claim mentioned ViTs as part of the general set of artificial neural networks that do not meet these hypothetical requirements. Here we should emphasize that this hypothesis is now questioned by some cognitive neuroscientists (Scholte et al. 2025, Beyond binding: from modular to natural vision).
>
> We would like to point out that this perspective is also shared by computer scientists (Greff et al. 2020): “In the case of neural networks, the binding problem is not just a gap in understanding but rather characterizes a limitation of existing neural networks.”
>
> Finally, We would remove ref11 from the references and include the work of Roelfsema as it is more consistent with our claim.
>
> **Our work on ViTs ties into a large and ongoing discussion about the capabilities of brains vs ANNs in object-based perception. We believe with the additional references we can make our relevance much clearer.**
>
> ---
>
> ### **[W4.1, Q4] Weak Attention–Binding Correlation**
> We agree that the correlation in Figure 6 (r = 0.163/0.201) is modest, but it is consistently positive. However, such modest effects should be expected, after all attention does a lot more than binding. It is meaningful to plot attention weights against the estimated IsSameObject, as it reveals the model’s implicit belief about whether two patches belong to the same object.
>
> To address the reviewer’s suggestion, we’ve extended our analysis to compare attention weights against the true patch‑pair labels, and find that attention is significantly higher for same‑object pairs than for any category of different‑object pairs.
>
> **Table 1**: Mean self‑attention weights for different patch‑pair categories
> | Pair Type                              | Mean Attention Weight |
> |----------------------------------------|-----------------------:|
> | Same‑object pairs                      |                  0.033 |
> | Different‑object pairs                 |                  0.014 |
> | Different cars (same class)           |                  0.013 |
> | Different classes (car – boat)         |                  0.018 |
>
>
> ### **[W4.2, Q4] Ambiguous ablation method and marginal effects**
>
> We’d like to clarify that “binding” part $b^l(x)$ is computed as $b^{(l)}(k) = h^{(l)}(x)^⊤W$, where W is the trained weight matrix of the quadratic probe. Our reasoning for this formulation is given in Lines 178–180.
> The ablation yields only modest performance gains (or losses), reflecting a relatively small effect size; we will explicitly note this as a limitation in the camera‑ready version’s limitations section.
>
> ---
>
> ### **[Q5]  Baseline definition in Figure 3**
> The baseline corresponds to the accuracy of always predicting "different", which accounts for the class imbalance in the dataset.
>
> ---
>
> ### **[Q6] Extraction of binding vectors from token embeddings**
> We obtain the binding component $b_a$ by projecting $h_a$ onto the quadratic probe’s weight matrix $W$:
> $b_a = h_a^\top W,\quad f_a = h_a - b_a.$ Although this projection is an approximation, since the “true” binding vector was computed by subtracting embeddings of identical objects (Lines 231–237; Fig. 4), it performs well in practice, as the probe was trained on large, natural-image datasets rich in similar-looking objects that can only be distinguished by binding signals (Lines 178-180).
>
> ---
> ### **Minor concerns**
> We thank the reviewer for their careful review. We will correct the LaTeX/formatting issues and restructure the section as suggested to improve clarity and flow.
>
> ---
>
> We believe that our responses have substantially strengthened the contributions of our work and are thankful for the reviewer's feedback.
>
> **Best regards,**
> *Authors*

---

> > ### Comment · Reviewer_x2fh · 2025-08-05
> >
> > Thank you for the response to all the reviews. I appreciate the additional experiments on both additional models and cross-layer binding. The rebuttal addresses my points and I am happy to raise my rating.

---

> > > ### Author Response · Authors · 2025-08-06
> > > **Thank you!**
> > >
> > > We're pleased that our additional experiments on models with different training objectives and cross-layer binding effectively addressed your concerns. Thank you for your constructive feedback throughout the review process, especially your comments on cross-layer binding, which have greatly improved our work.

---

### Official Review · Reviewer_43Cy · 2025-07-23

**Clarity:** 4
**Significance:** 2
**Originality:** 3
**Rating:** 5
**Confidence:** 4

**Summary:**

The authors study whether object binding - the ability of representations to bind low-level visual features into object level representations - appears in large self-supervised vision transformers. The authors propose a simpler definition of object binding, object features being grouped together, and propose a proxy task isSameObject to verify this definition in DINOv2 representations. In order to do so, the authors train various decoder probes on pairwise token features to classify whether two tokens (patches for vision transformers) encode features about the same object. The authors' experiments show that they are able to reliably perform this classifaction, especially with non-linear probes - which is an indicator that token level features encode object binding. They also show that early layers in ViTs bind object level information more while later layers bind class level information more. The authors also show that the isSameObject feature representations are low dimensional. Lastly, they show that isSameObject scores have a positive correlation with the attention weights, and hence are used in informing the model decisions.

**Questions:**

Please see weaknesses above

**Ethical Concerns:**

["NO or VERY MINOR ethics concerns only"]

**Final Justification:**

The additional pre-training objectives studied in the rebuttal improve the strength of the paper's central claim, so I have increased my score from Borderline to Accept. But I think the authors should clearly make a note about the different aspects of the binding problem in the paper itself, and mention how compositionality was not specifically explored in this paper and is left for future work. I also went through the notes and rebuttals from the other reviews, and after reading them my original comments on the strengths of the paper's key problem and solution remain, while the additional results reported alleviate the issues around lack of experimental coverage. The problem, approach and results are enough to justify an acceptance.

**Limitations:**

yes

**Quality:**

3

**Strengths And Weaknesses:**

# Strengths

1. Solving object binding has been a key concern in representation learning, and learning object-part heirarchies is a core problem in computer vision. Past research has focussed on building architectural inductive biases focussed on learning these (e.g. Slot Attention mentioned by authors, Capsule Networks). But understanding whether training objective based inductive biases enable the learning of object binding is a very important question - as it will inform which direction to focus future research on learning such representations. Past research (e.g. Grounding inductive biases in natural images:invariance stems from variations in data, Bouchacourt et al; Do vision transformers see like convolutional neural networks?, Raghu et al) has already shown that learning based inductive biases often end up learning similar feature properties as architecture based biases. This paper does a good job at establishing this for the isSameObject property, which is relevant for assessing object binding in learned representations (although not a 1:1 proxy, see weaknesses below).


# Weaknesses

1. I think the major limitation of the paper is that the all experiments are conducted on vision transformers with one particular self-supervised training objective (i.e. DINOv2) whereas the scope of the paper based on the title *Large Pretrained Vision Transformers* is much larger. In this sense just looking at one pre-training loss objective is too limited of a scope to draw conclusions about large pretrained ViTs in general. Furthermore, it seems like all experiments were done on the ViT-Base model, and there is no comparison on effect of model size on object binding properties either. The paper would be much more robust if these additional factors were studied.
2. Why can object binding only occur through inter-token interactions? I am referring to this claim by the authors:

> [L132-135] Object binding can only occur through token-token interactions, so ultimately, activation patterns useful for binding must inform self-attention, which takes a quadratic, pairwise form. Therefore, if ViTs indeed perform any form of object binding, we expect to observe a pairwise token-level representation that indicates whether two patches belong to the same object, which we term IsSameObject

Each token's features in successive layers goes through a multi-head attention where intra-token interactions are also happening (e.g. Key Value dot product, Scaling value with attention score). Saying only token-token interactions can lead to object binding is not entirely correct for ViT architectures.

3. I do not agree with the central claim "Since object binding is precisely the ability to group an object’s features together, decoding Is-SameObject reliably from ViT patch embeddings would provide direct evidence of object binding". This is incorrect. According to Greff et al "(Binding is) the capacity to acquire a compositional understanding of the world in terms of symbol-like entities (like objects)". So, binding is the ability to both group an object's features together (concept level features), and to reliably decode them while disentangling other objects' features to perform downstream tasks (compositionality). This corresponds to the third hypotheses about how IsSameObject may be encoded in model activations - "The signal is stored in a few specialized dimensions versus distributed across many dimensions". In the isSameObject task, the distributed quadratic probe consistently outperforms the diagonal quadratic probe, and heavily outperforms the linear probe. To me this is an indicator that the second part of the binding condition is not being satisfied in the DINOv2 representations.

---

> ### Author Rebuttal · Authors · 2025-07-31
>
> # Response to reviewer 43Cy(1/1)
>
> Dear Reviewer 43Cy,
>
> We thank the reviewer for recognizing our work's contribution in showing that training objective–based inductive biases can drive object binding independent of architectural design. We agree with their feedback and are delighted that we were able to run the suggested experiments (across different objectives and models) and that they substantially improved our paper.
>
> Below, we address their comments point by point.
>
> ---
>
> ### **[W1, Q1] Limited verification across pretraining objectives**
> We have now extended our analysis beyond DINOv2 to include CLIP, MAE, and supervised-only trained ViTs. (Some limited experiments with CLIP and DINO model sizes were already included in the supplementary: Lines 471-478, Appendix Figure 7). To ensure fair comparisons, we now standardized the effective patch coverage by resizing inputs so that all models process the same spatial patch divisions. This ensures a shared baseline (72.6%) across all models (corresponding to the accuracy of always predicting "different," which accounts for the class imbalance in the dataset).
>
> Here are our results:
>
> **Table: Probe accuracy on the *IsSameObject* task across ViTs**
> | **Model**                  | **Highest Probe Accuracy (%)** | **Improvement Over Baseline** | **Layer at Highest Accuracy (Normalized. 0=first layer, 1= last layer)** |
> |:-------------------------- |:-----------------------:|:----------------------------:|:--------------------------:|
> | **DINOv2‑Small**           | 86.7 %                  | + 14.1 pp                    | 1.00                       |
> | **DINOv2‑Base**            | 87.5 %                  | + 14.9 pp                    | 0.82                       |
> | **DINOv2‑Large**           | 90.2 %                  | + 17.6 pp                    | 0.78                       |
> | **DINOv2‑Giant**           | 88.8 %                  | + 16.2 pp                    | 0.77                       |
> | **CLIP (ViT‑Large)**       | 84.2 %                  | + 11.6 pp                    | 0.39                       |
> | **MAE (ViT‑Large)**        | 82.9 %                  | + 10.3 pp                    | 0.65                       |
> | **Supervised (ViT‑Large)** | 76.3 %                  | + 3.7 pp                     | 0.13                       |
> This now allows us to produce much more interesting insights:
> - **Pretraining objectives**: CLIP, MAE, and DINO all demonstrate object binding capabilities, supporting our broader claims about *Large Pretrained Vision Transformers* beyond DINOv2 alone. We also show that supervised ImageNet classification yields poor object‑binding performance, indicating that binding is an acquired ability under specific pretraining objectives rather than being a universal property of all vision models.
> - **Model size**: Larger models maintain object binding effectiveness, but the optimal layer for object binding shifts earlier in the network (lower normalized layer index), with performance plateauing in later layers (see supplementary Lines 471-478, Appendix Figure 7). This suggests that increasing model scale makes object binding emerge earlier, and deeper layers become redundant for binding.
>
> We offer explanations of why these training regimes work:
> - **DINO**: As noted in Lines 186–189, the DINO loss drives the model toward object-level features by enforcing consistency between augmented views, since each view contains the same objects and the model must align their representations; this objective has been shown to encourage class and object specific features (Caron et al. ICCV 2021), as the reviewer points out.
> - **CLIP**: By aligning images with text captions, CLIP effectively assigns each object a symbolic label (e.g., “the cat”), which can act like a pointer that pulls together all patches of that object. This supervision likely encourages patches from the same object to cluster in feature space. Future work could track object‑binding strength across CLIP training checkpoints to validate this hypothesis.
> - **MAE**: The masked autoencoder objective requires the model to reconstruct a missing patch from its surroundings. When the masked patch sits between multiple objects, correctly predicting its content forces the model to infer which object it belongs to, thus promoting the grouping of patches from the same object.
>
> **Our findings thus produce a much wider coverage of ViTs and we provide an understanding of potential reasons for why binding emerges.**
>
> ---
>
> ### **[W2] Object binding through intra-token interactions**
> As the reviewer correctly points out, multi‐head attention involves both intra‑token operations (the linear projections that mix channels within each token) and inter‑token operations (the dot‑product and weighted summation that relate different tokens). In our work, we define object binding strictly at the level of patch tokens, i.e. whether separate tokens group together to form a single object. Under this definition, no binding can occur within a single token alone, so intra‑token projections cannot contribute to the effect we study.
>
> An alternative notion–binding different attributes within one token, such as “red” and “square”, would require us to decompose a patch’s embedding into separate feature channels. We appreciate that investigating this within‑token compositionality is an important direction, but it falls outside the scope of our current study because it is likely to use rather different mechanisms (for example, dealing with a relatively small set of tokens with a large number of intra-token feature channels).
>
> ---
>
> ### **[W3] Lack of evidence of compositionality**
> We appreciate the reviewer's comment on the importance of binding in functionally approaching human level generalization in neural networks. The reviewer references a seminal work (Greff et al. 2020), which defines binding as the ability to dynamically and flexibly bind information that is distributed throughout the network, they also investigate its *functional aspects*—segregation, representation, and composition—without implying that all of them are required by the definition itself. We agree these are useful perspectives, but note that the literature *does not universally require compositionality as part of the core definition of binding*.
>
> The reviewer points out, while we clearly demonstrate (1) ViTs’ ability to group an object’s patches (i.e., segregation or parsing), we do not show (2) that those grouped features are disentangled into a truly compositional representation usable by downstream tasks. We appreciate this clearer framing and believe our results offer preliminary evidence of such compositional structure, yet we agree that a deeper exploration of compositionality lies ahead as future work.
>
> To clarify,
> - The distributed quadratic probe’s superior performance demonstrates that the *IsSameObject* signal spans many channels rather than being confined to a handful of specialized ones. While the reviewer interprets this distribution as evidence against compositionality, a distributed code can still be fully compositional and accessible to downstream tasks. Consider a signal that is intrinsically one‑dimensional but oriented along a rotated axis: when expressed in the original coordinate system, its nonzero components appear across multiple dimensions, yet its underlying structure remains compact.
> - We distinguish between distributed representation and low intrinsic dimensionality of the *IsSameObject* space:
>     - **Distributed representation** means the model uses several activation dimensions (channels) to encode whether two patches belong to the same object.
>     - **Low intrinsic dimensionality** means that, despite this distribution, all those activations lie on a simple, low‑dimensional manifold or subspace—so the binding information itself is inherently compact. We demonstrate this low‑dimensional clustering in Lines 231–237.
> - We acknowledge that a deeper investigation of compositionality in downstream tasks remains an important direction for future work. The empirical evidence for the compositionality of the feature representations is accumulating (Amir et al., Deep ViT Features as Dense Visual Descriptors, ECCVW 2022; Darcet et al., Vision Transformers Need Registers, 2023).
>
> We believe that our responses have substantially strengthened the contributions of our work and are thankful for the reviewer's feedback.
>
> **Best regards,**
> *Authors*

---

> > ### Comment · Reviewer_43Cy · 2025-08-06
> >
> > Thank you for the additional ablations with different pre training objectives. With regards to binding and compositionality, I think the note from the rebuttal should be worked into the paper, perhaps in the discussion or related works section. I am happy to raise my score based on the rebuttal.

---

> > > ### Author Response · Authors · 2025-08-06
> > > **Thank you!**
> > >
> > > We're glad our additional experiments addressed your concerns. We'll integrate the clarification on binding and compositionality into the discussion/related work section as suggested. Thank you for your valuable feedback that has strengthened our paper!

---

### Comment · Area_Chair_xbNT · 2025-08-03
**Please discuss the paper asap**

Dear reviewers,

Now the rebuttal is available. Thanks 4Wjo for starting the discussion. Others, please discuss with authors and among reviewers asap.

Please try to come to a consensus on the key issues even though the rating can be different. Please feel free to let me know how I can help.

Best,

Your AC

---

### Decision · Program_Chairs · 2025-09-17

**Decision:**

Accept (spotlight)

**Comment:**

This paper investigates whether object binding, the cognitive ability to group features into coherent object representations, emerges spontaneously in pretrained Vision Transformers (ViTs). Through a series of carefully designed probing experiments centered on a novel `IsSameObject` task, the authors provide compelling evidence that this capability is indeed present, develops through the model's hierarchy, is encoded in a low-dimensional subspace, and is functionally relevant to the model's pretraining objective. The paper received strong support from all reviewers, who converged on a clear consensus for acceptance after a highly effective rebuttal period.

The paper's primary strength lies in its novel and well-motivated research question, bridging the fields of cognitive science and deep learning interpretability. Reviewers commended the work for its clear setup, rigorous methodology (including probing, ablation studies, and correlation with attention), and the significance of its central inquiry. By moving beyond architectural solutions like slot-based attention, the paper provides valuable insights into how fundamental cognitive functions can arise from inductive biases in self-supervised learning objectives.

The most significant initial concern, shared unanimously by the reviewers, was the limited scope of the experiments. The paper's claims about "Large Pretrained Vision Transformers" were not fully supported by an analysis that focused almost exclusively on the DINOv2 pretraining objective. The authors comprehensively addressed this limitation during the rebuttal by conducting a substantial new set of experiments across a wider range of pretraining paradigms, including CLIP, MAE, and supervised-only ViTs. These new results not only generalized the original findings—showing that object binding emerges in multiple self-supervised frameworks—but also provided richer insights, notably demonstrating that standard supervised classification on ImageNet does *not* lead to strong binding capabilities. This experimental expansion was highly convincing and led all reviewers to confirm or raise their scores, solidifying the consensus for acceptance.

Beyond the expanded model scope, the authors also provided effective clarifications on several other points raised during the review, including the precise definition of binding versus compositionality (Reviewer 43Cy), the potential for cross-layer binding mechanisms (Reviewer x2fh), and the interpretation of specific figures and ablations (Reviewers gvWE, 4Wjo). The discussion was constructive and significantly improved the final manuscript.

In short, the resulting paper is a strong, well-supported, and insightful contribution. It addresses a fundamental question at the intersection of AI and cognitive science with a novel and effective experimental paradigm. The work is of high interest to researchers in model interpretability, self-supervised learning, and computational neuroscience. It presents a novel finding, supported by rigorous and now much broader evidence, and successfully navigated the peer-review process to become a significantly stronger manuscript.